# Zebrafish embryonic explants undergo genetically encoded self-assembly

**Alexandra Schauer†, Diana Pinheiro†, Robert Hauschild, Carl-Philipp Heisenberg***

IST Austria, Klosterneuburg, Austria

**Abstract** Embryonic stem cell cultures are thought to self-organize into embryoid bodies, able to undergo symmetry-breaking, germ layer specification and even morphogenesis. Yet, it is unclear how to reconcile this remarkable self-organization capacity with classical experiments demonstrating key roles for extrinsic biases by maternal factors and/or extraembryonic tissues in embryogenesis. Here, we show that zebrafish embryonic tissue explants, prepared prior to germ layer induction and lacking extraembryonic tissues, can specify all germ layers and form a seemingly complete mesendoderm anlage. Importantly, explant organization requires polarized inheritance of maternal factors from dorsal-marginal regions of the blastoderm. Moreover, induction of endoderm and head-mesoderm, which require peak Nodal-signaling levels, is highly variable in explants, reminiscent of embryos with reduced Nodal signals from the extraembryonic tissues. Together, these data suggest that zebrafish explants do not undergo *bona fide* self-organization, but rather display features of genetically encoded self-assembly, where intrinsic genetic programs control the emergence of order.

**\*For correspondence:**
heisenberg@ist.ac.at

†These authors contributed equally to this work

**Competing interests:** The authors declare that no competing interests exist.

**Reviewing editor:** Ashley Bruce,

## Introduction

Embryonic development entails both progressive specification of cells into distinct cell fates and their respective allocation into different domains within the embryo. Key signaling molecules, also termed morphogens, play critical roles in eliciting both the genetic programs underlying cell fate specification and the morphogenetic cell properties required to shape the developing embryo (*Gilmour et al., 2017*; *Heisenberg and Solnica-Krezel, 2008*; *Pinheiro and Heisenberg, 2020*). While amphibians critically rely on maternal factors for precise spatiotemporal activation of morphogen signaling during early embryonic development, mammalian embryos mostly depend on reciprocal interactions between embryonic and extraembryonic tissues (*Mummery and Chuva de Sousa Lopes SM, 2013*; *Heasman, 2006*; *Langdon and Mullins, 2011*; *Stern and Downs, 2012*; *Tam and Loebel, 2007*). Interestingly, zebrafish embryos combine features characteristic of both amphibian and mammalian development. Here, maternal factors initially deposited within the oocyte (*Langdon and Mullins, 2011*; *Marlow, 2020*) and signals from extraembryonic tissues, namely the yolk cell and its yolk syncytial layer (YSL), play important roles in germ layer induction and patterning (*Carvalho and Heisenberg, 2010*; *Chen and Kimelman, 2000*; *Erter et al., 1998*; *Fan et al., 2007*; *Gagnon et al., 2018*; *Hong et al., 2011*; *Mizuno et al., 1999*; *Mizuno et al., 1996*; *Ober and Schulte-Merker, 1999*; *Schier and Talbot, 2005*; *van Boxtel et al., 2015*; *Veil et al., 2018*; *Xu et al., 2012*). Nodal/TGFβ ligands, for instance, key signals involved in mesoderm and endoderm (mesendoderm) specification and patterning, are initially expressed under maternal control and then secreted by the extraembryonic YSL to pattern the embryonic blastoderm (*Chen and Schier, 2002*; *Dougan et al., 2003*; *Erter et al., 1998*; *Fan et al., 2007*; *Fauny et al., 2009*; *Feldman et al., 1998*; *Gagnon et al., 2018*; *Gore et al., 2005*; *Gritsman et al., 2000*; *Gritsman et al., 1999*; *Hong et al., 2011*; *Rebagliati et al., 1998*; *Rogers et al., 2017*; *Rogers and Müller, 2019*; *Thisse and Thisse, 2015*; *van Boxtel et al., 2015*; *Xu et al., 2012*).

Pioneering studies have suggested that both human and mouse embryonic stem cell (ES) colonies exhibit a remarkable capacity to self-organize *in vitro* and recapitulate features associated with early embryogenesis (*Baillie-Johnson et al., 2015*; *Beccari et al., 2018*; *Doetschman et al., 1985*; *Shahbazi et al., 2019*; *Simunovic et al., 2019*; *Simunovic and Brivanlou, 2017*; *ten Berge et al., 2008*; *Turner et al., 2017*; *Turner et al., 2014*; *van den Brink et al., 2014*; *van den Brink et al., 2020*; *Warmflash et al., 2014*). For instance, homogeneous stimulation of human ES colonies cultured on 2-dimensional circular substrates with BMP4 ligands is sufficient to reproducibly generate radially symmetric colonies, where cells at the outmost rim differentiate into extraembryonic tissue, while the remaining cells differentiate progressively into more internal rings composed of the different embryonic germ layers - endoderm, mesoderm and ectoderm (*Warmflash et al., 2014*). These apparent self-organizing processes are thought to rely on boundary sensing, community effects and the interplay between activators and inhibitors (*Chhabra et al., 2019*; *Etoc et al., 2016*; *Nemashkalo et al., 2017*; *Sasai, 2013*; *Xavier da Silveira Dos Santos and Liberali, 2019*). While these pioneering studies suggest that embryonic tissues might have some self-organizing capacity, classical studies have also established the dependency of embryonic development on both maternal factors and extraembryonic tissues, thereby raising key questions as to their specific contributions to robust embryonic development.

## Results

To address this question, we established an *ex vivo* system to culture zebrafish blastoderm tissue explants lacking the yolk cell and its associated YSL. For this, blastoderm explants were prepared from embryos at early cleavage stages (256 cell stage), before YSL formation or germ layer induction, and kept in serum-free medium until stage-matched control embryos had completed gastrulation (bud stage; *Figure 1A*). We found that these blastoderm explants initially rounded up in culture, but eventually became 'pear'-shaped by forming an extension (*Figure 1B,C*, *Video 1*). Interestingly, this extension began to form when stage-matched control embryos initiated body axis extension at shield stage (*Sepich et al., 2005*; *Solnica-Krezel and Sepich, 2012*), pointing at the possibility that these processes might be related. Contrary to intact embryos, however, blastoderm explants formed a large luminal cavity, reminiscent of a blastocoel (*Krens et al., 2017*), which was typically positioned at the opposite end of the explant extension (*Figure 1B*, *Figure 1—figure supplement 1A,B*, *Video 2*). Importantly, cells within explants appeared to normally complete their rounds of cleavages and divisions following explant preparation, as evidenced by the similar average diameter of explant and stage-matched embryo cells (*Figure 1—figure supplement 1C*). Moreover, we found that the onset of zygotic genome activation (ZGA), as monitored by the expression of the two zygotic genes *tbx16* and *chrd* (*Gagnon et al., 2018*; *Lee et al., 2013*), was similar in stage-matched embryos and blastoderm explants (*Figure 1—figure supplement 1D*). Finally, comparable morphological changes could be observed when blastoderm explants were prepared from earlier (64 and 128 cell-stage) or later (high stage) embryos (*Figure 1—figure supplement 2A–D*), suggesting that the exact time point of explant preparation during pre-gastrula stages has no decisive influence on explant morphogenesis.

To analyze how these distinct changes in explant morphology relate to the specification of different progenitor cell types therein, we analyzed the expression of various progenitor cell marker genes by whole mount *in situ* hybridization. Strikingly, we found distinct expression domains for all tested marker genes outlining ectoderm, neuroectoderm, endoderm and different types of mesoderm progenitor cell populations. While ectoderm (*gata2*) and neuroectoderm (*six3*) marker genes were expressed predominantly at the opposite end of the explant extended portion, the region where the cavity was also forming (*Figure 1D–G*, *Figure 1—figure supplement 1A,B,H*, *Figure 1—figure supplement 3A*), all mesendoderm and endoderm marker genes were exclusively found within or adjacent to the extension (*Figure 1D–G*, *Figure 1—figure supplement 1G–J*, *Figure 1—figure supplement 3A*). Specifically, we found that markers of posterior axial mesoderm (*ntl* and *flh*) were expressed in the central region of the extension, flanked by paraxial and lateral mesoderm (*papc*, *myoD*, *tbx6*) (*Figure 1D–G*, *Figure 1—figure supplement 1G–I*, *Figure 1—figure supplement 3A*). In contrast, anterior axial mesendoderm markers (*hgg* and *gsc*) were found to be expressed at variable positions along the extended region or directly adjacent to it (*Figure 1D–G*, *Figure 1—figure supplement 1G–I*, *Figure 1—figure supplement 3A*). Finally, sporadic endoderm progenitors

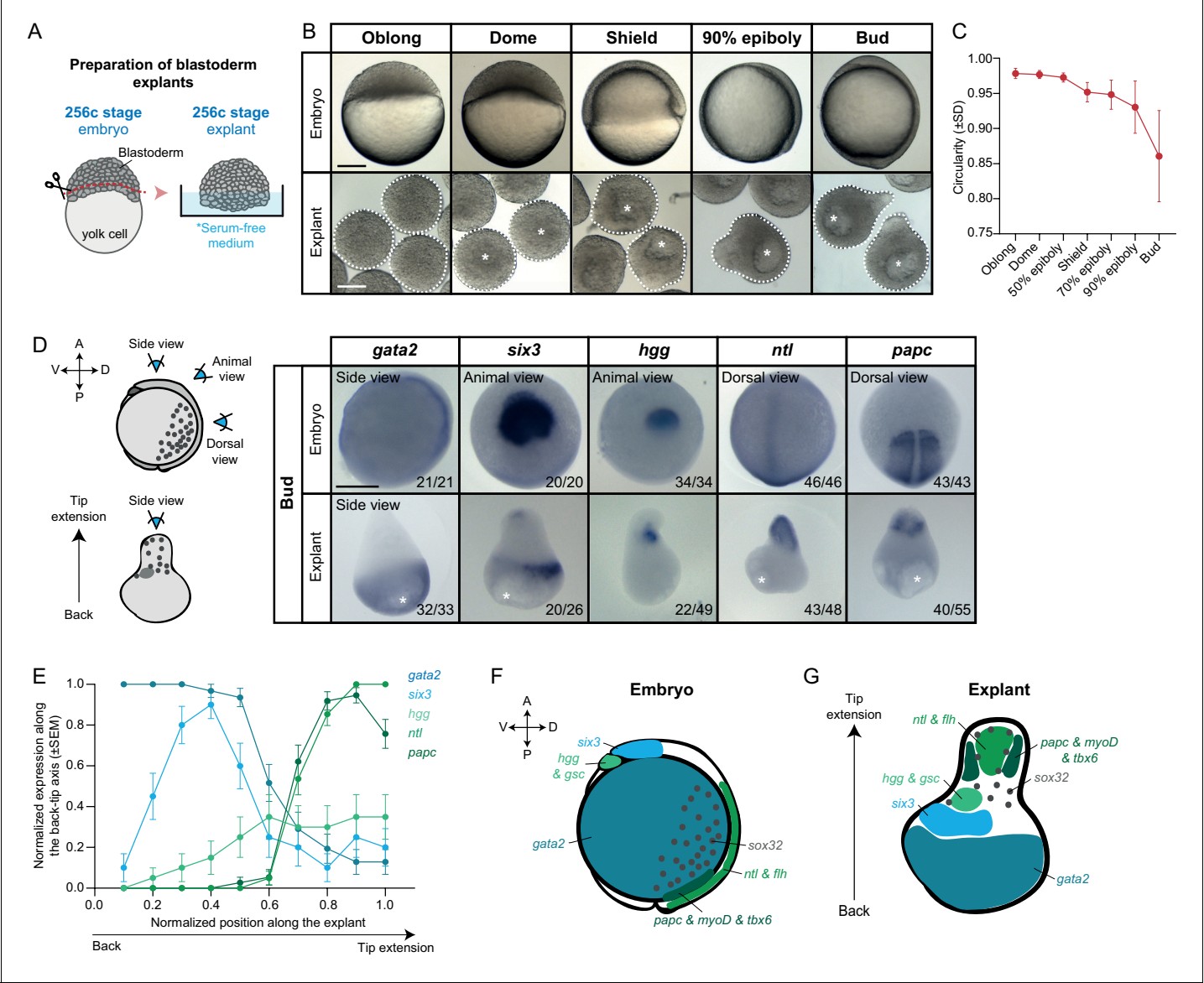

**Figure 1.** Mesendoderm specification and morphogenesis in the absence of the yolk cell and its associated YSL. (**A**) Schematic representation of the preparation method of blastoderm explants from 256 cell stage (256 c) stage embryos. (**B**) Bright-field single-plane images of stage-matched embryo and blastoderm explants from oblong to bud stage. The white dashed lines outline the shape of the explant. (**C**) Circularity of blastoderm explants from oblong to bud stage (oblong: n = 42, dome: n = 34, 50% epiboly: n = 35, shield: n = 40, 70% epiboly: n = 35, 90% epiboly: n = 36, bud: n = 38; N = 2). (**D**) Expression of ectoderm (*gata2*), neuroectoderm (*six3*) and mesendoderm (*hgg, ntl* and *papc*) marker genes as determined by whole mount *in situ* hybridization of bud stage embryos and blastoderm explants. Schematic representation of the different views for embryos and blastoderm explants is shown on the left. The proportion of embryos or blastoderm explants with a phenotype similar to the images shown is indicated in the lower right corner (*gata2*: embryos, n = 21, N = 4, explants, n = 33, N = 6; *six3*: embryos, n = 20, N = 3, explants, n = 26, N = 4; *hgg*: embryos, n = 34, N = 5, explants, n = 49, N = 5; *ntl*: embryos, n = 46, N = 4, explants, n = 48, N = 4; *papc*: embryos, n = 43, N = 4, explants, n = 55, N = 5). (**E**) Normalized expression domain of ectoderm (*gata2*: n = 31, N = 6), neuroectoderm (*six3*: n = 20, N = 4) and mesendoderm (*hgg*: n = 20, N = 5; *ntl*: n = 41, N = 4; *papc*: n = 37, N = 5) marker genes along the back-tip axis of bud stage blastoderm explants. (**F-G**) Schematic representation of ectoderm, neuroectoderm, mesendoderm and endoderm marker gene expression domains in intact embryos (**F**) and blastoderm explants (**G**). White asterisks denote the main luminal cavity in explants. Scale bars: 200 μm (**B, D**).

The online version of this article includes the following source data and figure supplement(s) for figure 1:

**Source data 1.** Mesendoderm specification and morphogenesis in the absence of the yolk cell and its associated YSL.

**Figure supplement 1.** Phenotypic characterization of blastoderm explants.

**Figure supplement 1—source data 1.** Phenotypic characterization of blastoderm explants.

**Figure supplement 2.** Morphogenesis of blastoderm explants prepared at 64 c, 128 c, 256 c or high stage.

*Figure 1 continued on next page*

*Figure 1 continued*

**Figure supplement 2—source data 1.** Morphogenesis of blastoderm explants prepared at 64 c, 128 c, 256 c and high stage.
**Figure supplement 3.** Relative spatial distribution of ectoderm and mesendoderm progenitor cell types in blastoderm explants.
**Figure supplement 4.** Blastoderm explants exhibit reduced or no YSL features.
**Figure supplement 4—source data 1.** Blastoderm explants exhibit reduced or no expression of YSL markers.

(*sox32*) could be found throughout the extension (*Figure 1G*, *Figure 1—figure supplement 1G,H,J*, *Figure 1—figure supplement 3A*). On the explant surface, much like in the intact embryo, we found that cells expressed elevated levels of Keratin 4, a marker of enveloping layer (EVL) cell differentiation (*Figure 1—figure supplement 1E,F*). Collectively, these findings suggest that the blastoderm explants contain all main germ layer progenitors also found in intact embryos and display a seemingly complete patterning of the mesendoderm anlage. They also indicate that, although explants undergo extensive morphogenesis, mesendoderm progenitors do not clearly internalize and, instead, organize into an extension (*Video 3*). This is highly reminiscent of the 'exogastrula' phenotype observed in embryos with defective mesendoderm internalization and, more recently, in mouse gastruloids (*Beccari et al., 2018*; *Branford and Yost, 2002*; *Turner et al., 2017*; *van den Brink et al., 2014*).

To ensure that all extraembryonic tissues were efficiently removed during explant preparation and that explants do not form a secondary YSL, we analyzed the expression of marker genes, normally present within the YSL in intact embryos, in blastoderm explants. We found that all of these genes were either not expressed in explants or showed only sporadic expression in small groups of blastoderm cells (*Figure 1—figure supplement 4A,B*). Moreover, we failed to detect any clearly recognizable syncytium in explants, a hallmark of the YSL in intact embryos (*Figure 1—figure supplement 4C*). Together, these data strongly support the notion that germ layer specification and mesendoderm patterning in blastoderm explants can occur in the absence of the extraembryonic yolk and YSL.

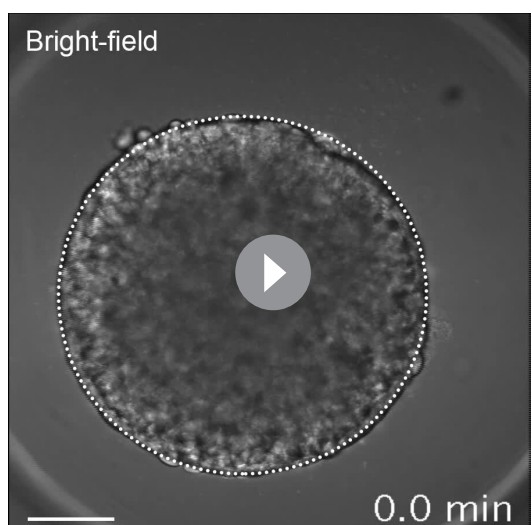

**Video 1.** Blastoderm explant morphogenesis in the absence of the yolk cell and its associated YSL. Bright-field time-lapse imaging of a blastoderm explant (cross-section) from late sphere to bud stage (n = 10, N = 3). The white dashed lines outline the initial and final shape of the explant. White asterisks denote the main luminal cavity. Time in min. Scale bar: 200 µm.
https://elifesciences.org/articles/55190#video1

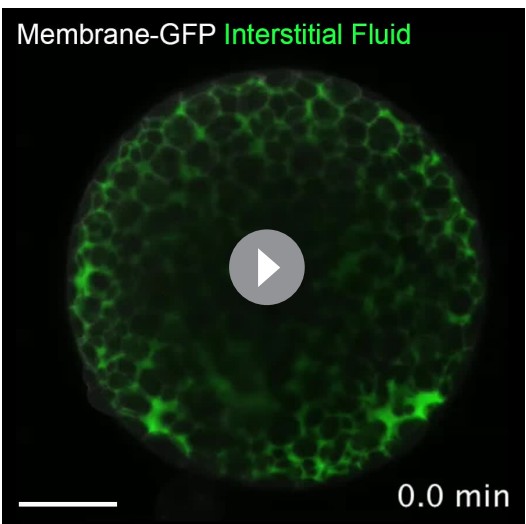

**Video 2.** Formation of the luminal cavity in blastoderm explants. High-resolution fluorescence time-lapse imaging of a blastoderm explant (cross-section) from oblong stage onwards (n = 9, N = 2). Interstitial fluid (green) is marked by dextran Alexa Fluor 647 and the cell membranes (white) are marked by Membrane-GFP. White asterisks denote the two luminal cavities. Time in min. Scale bar: 100 µm.
https://elifesciences.org/articles/55190#video2

To understand how mesendoderm tissues are formed within explants, we first analyzed whether Nodal ligands, critical for mesendoderm specification and patterning *in vivo* (*Erter et al., 1998*; *Feldman et al., 1998*; *Gritsman et al., 2000*; *Gritsman et al., 1999*; *Rebagliati et al., 1998*), are also expressed in explants. We observed expression of both Nodal ligands *sqt* and *cyc* in blastoderm explants (*Figure 2—figure supplement 1A,B*), consistent with previous observations in intact embryos (*Erter et al., 1998*; *Fan et al., 2007*; *Feldman et al., 1998*; *Rebagliati et al., 1998*; *van Boxtel et al., 2015*). In addition to this, we detected nuclear localization of phosphorylated SMAD2/3 (pSMAD2/3), a well-established readout of Nodal/TGFβ signaling activation (*Economou and Hill, 2020*; *Schier, 2009*; *van Boxtel et al., 2015*; *van Boxtel et al., 2018*), in cells located close to the wounding site of explants (*Figure 2A*, *Figure 2—figure supplement 1C,D*), indicative of local Nodal signaling activation. A comparison of the spatiotemporal dynamics of Nodal signaling establishment between explants and intact embryos further revealed that, while blastoderm explants showed initially fewer and less bright pSMAD2/3 positive nuclei than intact embryos, both parameters continued to increase in explants until stage-matched embryos had reached shield stage (*Figure 2A–C*, *Figure 2—figure supplement 1C–J*). This differs from the temporal dynamics of Nodal signaling in intact embryos, where nuclear accumulation of pSMAD2/3 peaks already by 50% epiboly (*Figure 2A–C*, *Figure 2—figure supplement 1C–J*). Notably, the wounding site of blastoderm explants, where Nodal signaling was activated corresponds to the former blastoderm margin where explants were removed from the yolk cell (*Figure 2D*, *Figure 2—figure supplement 1E–J*). Given that Nodal signaling in intact embryos is initiated in the blastoderm margin (*Dubrulle et al., 2015*; *Harvey and Smith, 2009*; *Schier, 2009*; *van Boxtel et al., 2015*), this points at the intriguing possibility that Nodal signaling is active in similar cells in embryos and explants. This notion was further substantiated by our observation that cells close to the wounding site in explants display only very limited dispersal during explant maturation (*Figure 2—figure supplement 2A–D*). Moreover, similar to intact embryos, a gradient of BMP signaling was also detectable in blastoderm explants (*Figure 2—figure supplement 2E,F*), suggesting that explants might also use a BMP signaling gradient for patterning along their dorsoventral axis.

To test whether and when Nodal signaling is required for mesendoderm induction and morphogenesis in blastoderm explants, we prepared explants from maternal-zygotic (MZ) mutants of the Nodal co-receptor Oep (MZ*oep*), previously shown to be essential for Nodal signaling and proper mesendoderm induction during embryogenesis (*Gritsman et al., 1999*). We found that blastoderm explants obtained from MZ*oep* mutants failed to extend, indicative of strongly reduced mesendoderm formation (*Figure 2E–G*). This suggests that, similar to the situation in intact embryos, proper Nodal signaling is critical for mesendoderm induction and morphogenesis in blastoderm explants. To determine when Nodal signaling is required for mesendoderm induction and morphogenesis, we cultured explants in the presence of the selective and reversible Nodal inhibitor SB-505124 (50 μM) (*DaCosta Byfield et al., 2004*; *Fan et al., 2007*; *Hagos and Dougan, 2007*) for different time-periods during explant maturation. We found that blocking Nodal signaling from 256 cell to shield stage was sufficient to completely abolish mesendoderm induction and explant extension (*Figure 2E–G*, *Figure 2—figure supplement 3A–D*), while blocking Nodal signaling after shield stage had no strong impact on these processes

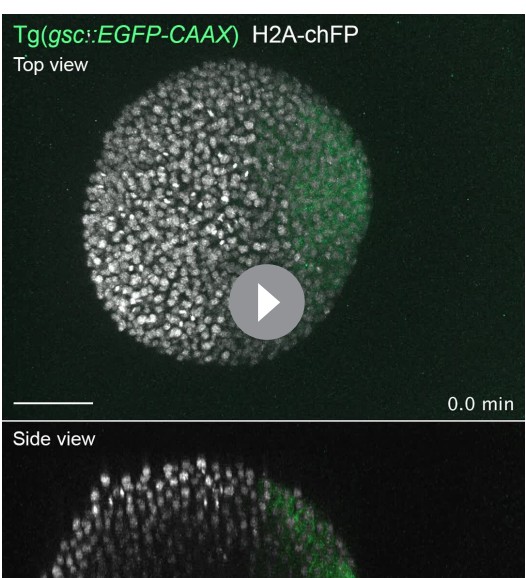

**Video 3.** Mesendoderm progenitor cells organize into an extension without undergoing clear internalization movements. High-resolution fluorescence time-lapse imaging of a blastoderm explant from late 50% epiboly onwards (n = 14, N = 14). The cell nuclei (white) are marked by H2A-chFP and the mesendoderm tissue (green) by expression of *gsc::GFP-CAAX*. The explant is shown both from the top and from a side view along the extension axis. Time in min. Scale bar: 100 μm.
https://elifesciences.org/articles/55190#video3

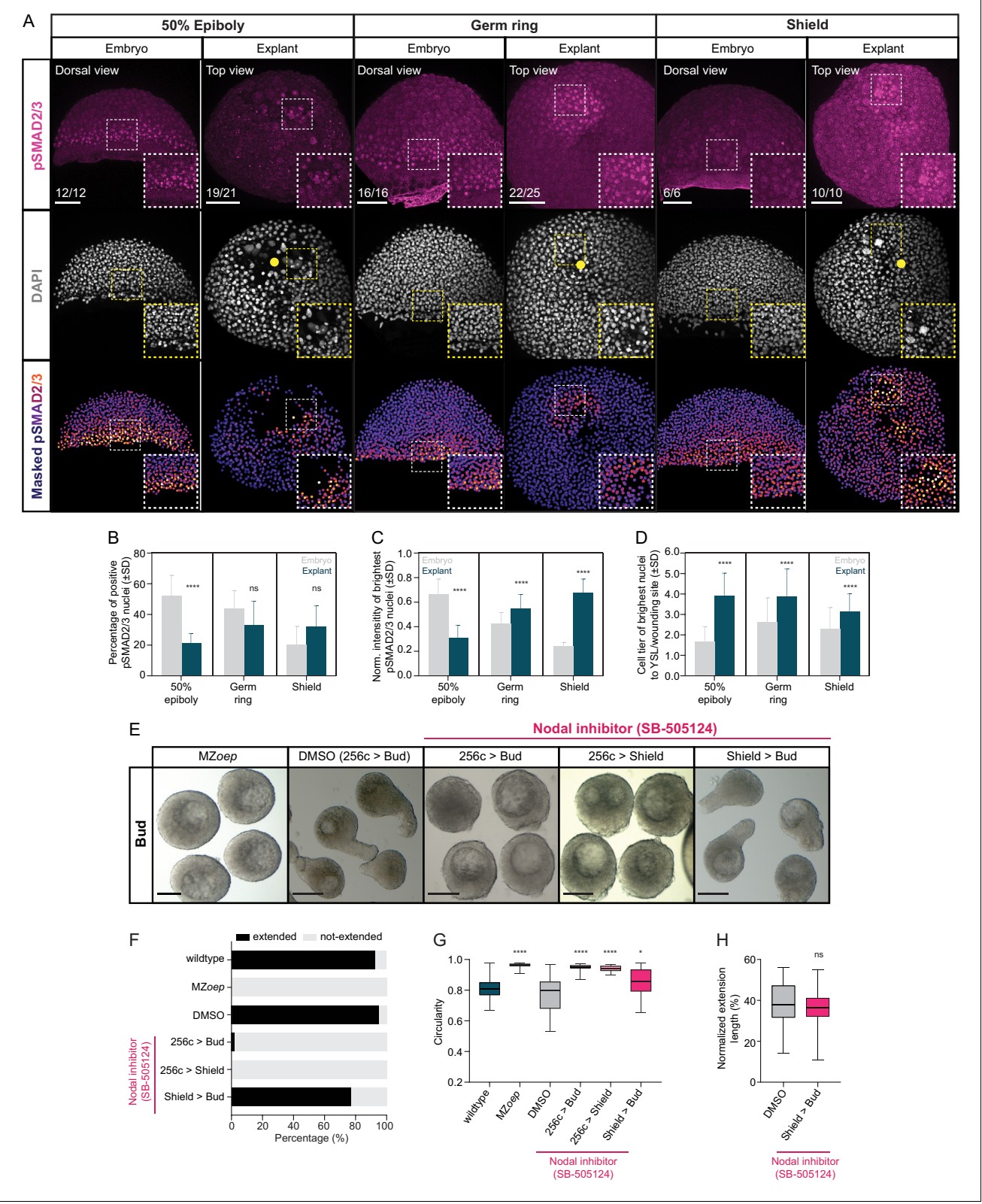

**Figure 2.** Nodal signaling is active in blastoderm explants and is required for tissue extension. (**A**) High-resolution fluorescence images of stage-matched embryos (dorsal view) and blastoderm explants (top view) at 50% epiboly, germ ring and shield stage stained for both pSMAD2/3 (pink) and DAPI (grey). Nuclear pSMAD2/3 is color-coded using a fire lookup table (highest intensities in yellow) and was masked based on the DAPI signal. Insets are zoom-in images of the highlighted regions (dashed boxes). Yellow circles denote the wounding site in explants. The proportion of embryos and

*Figure 2 continued on next page*

*Figure 2 continued*

explants with a phenotype similar to the images shown is indicated in the lower left corner (50% epiboly: embryos, n = 12, N = 4, explants, n = 21, N = 4; germ ring: embryos, n = 16, N = 4, explants, n = 25, N = 4 and shield: embryos, n = 6, N = 3, explants, n = 10, N = 3). (B) Percentage of pSMAD2/3 positive nuclei in stage-matched embryos and blastoderm explants at 50% epiboly (embryos: n = 7, N = 3; explants: n = 10, N = 3), germ ring (embryos: n = 10, N = 4; explants: n = 10, N = 4) and shield stage (embryos: n = 6, N = 3; explants: n = 8, N = 3). ****p<0.0001, ns, not significant (ANOVA test). (C) Normalized intensity of the brightest pSMAD2/3 nuclei (for details see Materials and methods) in stage-matched embryos and blastoderm explants at 50% epiboly (embryos: n = 7, N = 3; explants: n = 10, N = 3), germ ring (embryos: n = 10, N = 4; explants: n = 10, N = 4) and shield stage (embryos: n = 6, N = 3; explants: n = 8, N = 3). ****p<0.0001 (Kruskal-Wallis test). (D) Normalized distance, expressed as cell tiers, of the brightest pSMAD2/3 nuclei (for details see Materials and methods) from the YSL or wounding site in stage-matched embryos and blastoderm explants at 50% epiboly (embryos: n = 7, N = 3; explants: n = 10, N = 3), germ ring (embryos: n = 10, N = 4; explants: n = 10, N = 4) and shield stage (embryos: n = 6, N = 3; explants: n = 8, N = 3). ****p<0.0001 (Kruskal-Wallis test). (E) Bright-field single-plane images of bud stage MZ*oep* (n = 40, N = 3), DMSO (treated from 256 c to bud, n = 81, N = 5) or Nodal inhibitor (SB-505124)-treated blastoderm explants (treated from 256 c to bud, n = 49, N = 4; 256 c to shield, n = 38, N = 3; shield to bud, n = 43, N = 4). (F) Percentage of extended or not-extended wildtype (n = 26, N = 3), MZ*oep* (n = 40, N = 3), DMSO (treated from 256 c to bud, n = 81, N = 5) and Nodal inhibitor (SB-505124)-treated blastoderm explants (treated from 256 c to bud, n = 49, N = 4; 256 c to shield, n = 38, N = 3; shield to bud, n = 43, N = 4) at bud stage. (G) Circularity of bud stage wildtype (n = 26, N = 3), MZ*oep* (n = 40, N = 3), DMSO (treated from 256 c to bud, n = 81, N = 5) and Nodal inhibitor (SB-505124)-treated blastoderm explants (treated from 256 c to bud, n = 49, N = 4; 256 c to shield, n = 38, N = 3; shield to bud, n = 43, N = 4). ****p<0.0001, *p=0.0112 (Kruskal-Wallis test). (H) Normalized extension length of extended DMSO (treated from 256 c to bud, n = 77, N = 5) and Nodal inhibitor (SB-505124)-treated blastoderm explants (treated from shield to bud, n = 33, N = 4) at bud stage. ns, not significant (Unpaired *t* test). Scale bars: 100 µm (A), 200 µm (E).

The online version of this article includes the following source data and figure supplement(s) for figure 2:

**Source data 1.** Nodal signaling is active in blastoderm explants and is required for tissue extension.
**Figure supplement 1.** Temporal and spatial dynamics of Nodal signaling in blastoderm explants.
**Figure supplement 1—source data 1.** Temporal and spatial dynamics of Nodal signaling in the intact embryo and blastoderm explants.
**Figure supplement 2.** Reduced cell mixing in blastoderm explants close to the wounding site.
**Figure supplement 2—source data 1.** Reduced cell mixing in blastoderm explants close to the wounding site.
**Figure supplement 3.** Nodal signaling is required for mesendoderm specification in blastoderm explants.
**Figure supplement 3—source data 1.** Nodal signaling is required for mesendoderm specification in blastoderm explants.

(*Figure 2E–H*, *Figure 2—figure supplement 3A–D*). This is highly reminiscent of the early requirement of Nodal signaling for mesendoderm induction and morphogenesis in intact embryos (*Gritsman et al., 1999*; *Hagos and Dougan, 2007*; *Pinheiro and Heisenberg, 2020*), suggesting that Nodal signaling functions similarly in embryos and blastoderm explants.

Nodal-dependent mesendoderm induction has previously been shown to promote convergence and extension movements and body axis elongation by inducing the non-canonical Wnt/Planar Cell Polarity (Wnt/PCP) signaling module (*Djiane et al., 2000*; *Gray et al., 2011*; *Tada and Heisenberg, 2012*; *Wallingford, 2012*; *Williams and Solnica-Krezel, 2019*). To analyze whether explant extension also relies on non-canonical Wnt/PCP signaling, we prepared explants from MZ mutant embryos for different components of the Wnt/PCP signaling module, which display strongly diminished body axis elongation *in vivo*. Blastoderm explants prepared from MZ mutants for the Wnt/PCP components *wnt11*, *wnt5b* and *fz7a/b* also exhibited reduced or no extension (*Figure 3A–D*). This suggests that explant extension, similar to body axis elongation in intact embryos, critically depends on non-canonical Wnt/PCP signaling. In line with this, we also found that during explant maturation, the extension length increased, and the width of the non-extended portion decreased (*Figure 3—figure supplement 1A–C*), and that explant extension was largely independent of cell proliferation (*Figure 3—figure supplement 1D–F*). This points at the intriguing possibility that explants, like intact embryos, might undergo some type of convergence and extension movements, independent of cell proliferation.

Our data, so far, suggest that mesendoderm specification and morphogenesis in explants relies on the activation of Nodal signaling around the wounding site. In intact embryos, Nodal ligand expression, and thus activation of Nodal signaling, at the blastoderm margin depends both on initial maternal activation of the transcriptional co-activator β-catenin (β-cat) and Nodal signals emanating from the YSL (*van Boxtel et al., 2015*; *Feldman et al., 1998*; *Gagnon et al., 2018*; *Hong et al., 2011*; *Langdon and Mullins, 2011*; *Marlow, 2020*; *Xu et al., 2012*). Since explants contain, at most, remnants of the YSL (*Figure 1—figure supplement 4A–C*), we tested whether Nodal signaling activation in this context might be triggered by β-cat. Indeed, analysis of nuclear localization of β-cat in explants shortly after preparation revealed elevated levels of nuclear β-cat in blastoderm cells

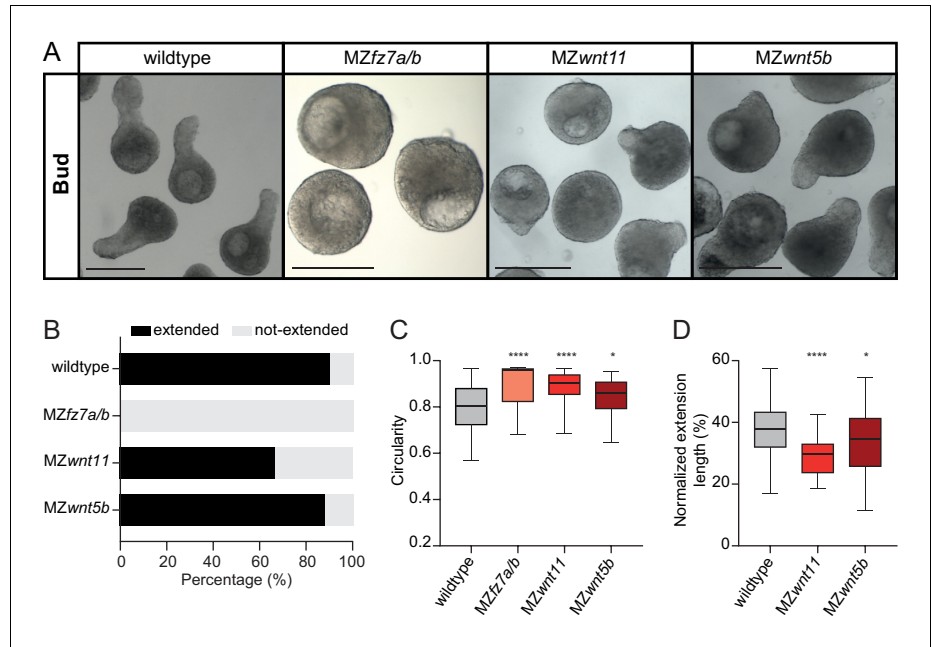

**Figure 3.** Non-canonical Wnt/PCP signaling is required for blastoderm explant morphogenesis. (**A**) Bright-field single-plane images of wildtype (n = 120, N = 8), MZ*fz7a/b* (n = 31, N = 3), MZ*wnt11* (n = 69, N = 5) and MZ*wnt5b* (n = 75, N = 4) blastoderm explants at bud stage. Scale bars: 200 µm. (**B**) Percentage of extended or not-extended wildtype (n = 120, N = 8), MZ*fz7a/b* (n = 31, N = 3), MZ*wnt11* (n = 69, N = 5) and MZ*wnt5b* (n = 75, N = 4) blastoderm explants at bud stage. (**C**) Circularity of wildtype (n = 120, N = 8), MZ*fz7a/b* (n = 31, N = 3), MZ*wnt11* (n = 69, N = 5) and MZ*wnt5b* (n = 75, N = 4) blastoderm explants at bud stage. ****$p<0.0001$, *$p=0.0424$ (Kruskal-Wallis test). (**D**) Normalized extension length of extended wildtype (n = 108, N = 8), MZ*wnt11* (n = 46, N = 5) and MZ*wnt5b* (n = 66, N = 4) blastoderm explants at bud stage. ****$p<0.0001$, *$p=0.0461$ (ANOVA test).

The online version of this article includes the following source data and figure supplement(s) for figure 3:

**Source data 1.** Non-canonical Wnt/PCP signaling is required for blastoderm explant morphogenesis.
**Figure supplement 1.** Cell proliferation is dispensable during blastoderm explant extension.
**Figure supplement 1—source data 1.** Cell proliferation is dispensable during blastoderm explant extension.

close to the wounding site (*Figure 4A–C*). This suggests that in explants, similar to intact embryos, β-cat nuclear accumulation might be responsible for Nodal signaling activation, and consequently, mesendoderm induction and morphogenesis.

Nuclear accumulation of β-cat is thought to be triggered by maternal factors, which are transported to the future dorsal side of the zebrafish embryo upon fertilization (*Langdon and Mullins, 2011*; *Marlow, 2020*). To determine whether this β-cat nuclear accumulation at the dorsal blastoderm margin is indeed responsible for Nodal signaling activation and mesendoderm induction in explants, we prepared explants from embryos depleted of maternal dorsal determinants (DD-removed) and analyzed the formation of the Nodal signaling domain therein. To this end, we mechanically removed the vegetal-most part of the zygote cortex, where these factors are thought to localize prior to transport to the future dorsal side of the embryo soon after fertilization (*Langdon and Mullins, 2011*; *Marlow, 2020*; *Mizuno et al., 1999*; *Ober and Schulte-Merker, 1999*). As expected, in blastoderm explants prepared from DD-removed embryos, a large portion (>50%) of which displayed a fully ventralized phenotype one day after fertilization (1 dpf), nuclear accumulation of β-cat was strongly diminished (*Figure 4—figure supplement 1A–D*). Interestingly, we found that explants originating from DD-removed embryos showed a clear reduction in Nodal signaling (*Figure 4D,E*, *Video 4*) and reduced expression of the pan-mesendodermal marker *sebox* (*Figure 4—figure supplement 1E–G*). Consistent with the notion that Nodal-mediated mesendoderm specification was reduced in DD-removed explants, we also found that the proportion of blastoderm explants forming an extension and the degree of explant extension were reduced

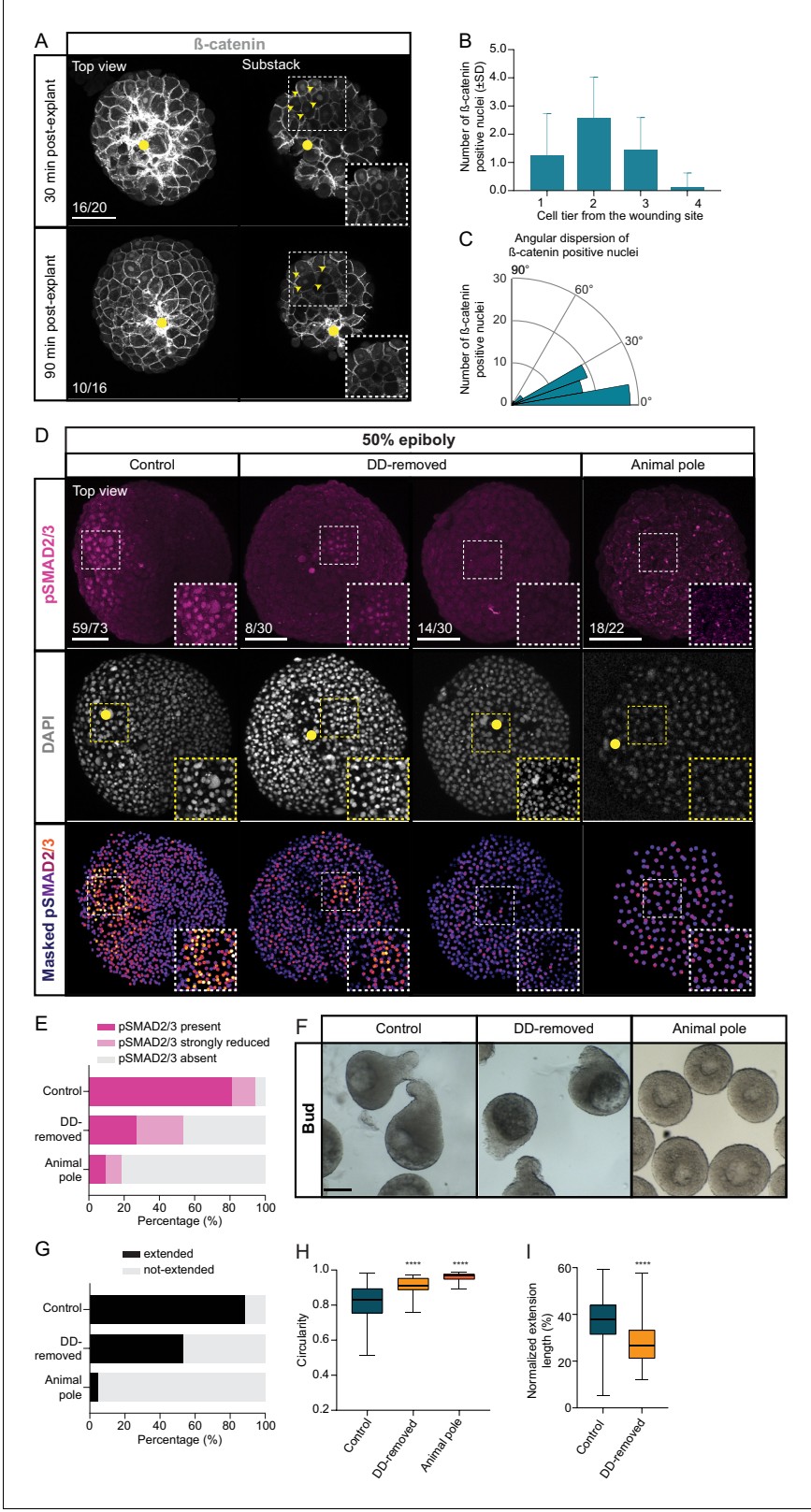

**Figure 4.** Maternal pre-patterning is required for Nodal signaling in blastoderm explants. (**A**) High-resolution fluorescence images of blastoderm explants 30 and 90 min post-explanting stained for both β-catenin (grey) and DAPI (not shown). Both full projection and substack top views are shown to facilitate simultaneous visualization of the wounding site (yellow circle) and nuclear accumulation of β-catenin (yellow arrowheads). Insets are zoom-in

*Figure 4 continued on next page*

*Figure 4 continued*

images of the highlighted regions (dashed boxes). The proportion of blastoderm explants with a phenotype similar to the images shown is indicated in the lower left corner (30 min: n = 20, N = 6; 90 min: n = 16, N = 4). (**B**) Number of β-catenin positive nuclei 30 min post-explant preparation as a function of the distance to the wounding site, expressed as cell tiers (n = 16, N = 6). (**C**) Angular dispersion of β-catenin positive nuclei 30 min post-explant preparation (for details see Materials and methods; n = 16, N = 6). (**D**) High-resolution fluorescence images of control, dorsal determinants-removed (DD-removed) and animal pole explants at 50% epiboly stained for both pSMAD2/3 (pink) and DAPI (grey). Nuclear pSMAD2/3 is color-coded using a fire lookup table (highest intensities in yellow) and was masked based on the DAPI signal. Insets are zoom-in images of the highlighted regions (dashed boxes) and the yellow circles denote the wounding site. The proportion of blastoderm explants with a phenotype similar to the images shown is indicated in the lower left corner (control: n = 73, N = 14; DD-removed: n = 30, N = 7; animal pole: n = 22, N = 6). (**E**) Percentage of control (n = 73, N = 14), DD-removed (n = 30, N = 7) and animal pole (n = 22, N = 6) explants showing a domain of pSMAD2/3 positive nuclei (present), a few sporadic pSMAD2/3 positive nuclei (strongly reduced) or no positive nuclei (absent) at 50% epiboly (see Materials and methods for additional details). (**F**) Bright-field single-plane images of control (n = 228, N = 15), DD-removed (n = 75, N = 8) and animal pole (n = 42, N = 5) explants at bud stage. Control explants partially correspond to explants shown in *Figure 1—figure supplement 2A–D*. (**G**) Percentage of extended or not-extended control (n = 228, N = 15), DD-removed (n = 75, N = 8) and animal pole (n = 42, N = 5) explants at bud stage. (**H**) Circularity of control (n = 228, N = 15), DD-removed (n = 75, N = 8) and animal pole (n = 42, N = 5) explants at bud stage. ****p<0.0001 (Kruskal-Wallis test). (**I**) Normalized extension length of extended control (n = 199, N = 15) and DD-removed (n = 40, N = 8) blastoderm explants at bud stage. ****p<0.0001 (Unpaired *t* test). Scale bars: 100 μm (**A,D**), 200 μm (**F**).

The online version of this article includes the following source data and figure supplement(s) for figure 4:

**Source data 1.** Maternal pre-patterning is required for Nodal signaling in blastoderm explants.
**Figure supplement 1.** Maternal pre-patterning is required for β-catenin nuclear localization.
**Figure supplement 1—source data 1.** Maternal pre-patterning is required for β-catenin nuclear localization.
**Figure supplement 2.** Loss of dorsal determinants is not sufficient to abolish Nodal signaling within the intact embryo.
**Figure supplement 2—source data 1.** Loss of dorsal determinants is not sufficient to abolish Nodal signaling in the intact embryo.
**Figure supplement 3.** Mesoderm specification in blastomere reaggregates.
**Figure supplement 3—source data 1.** Mesoderm specification in blastomere reaggregates.

---

(*Figure 4F–I*). Moreover, the extent of mesendoderm induction in DD-removed explants closely correlated with the degree of explant extension at bud stage (*Figure 4—figure supplement 1H*). This suggests that in blastoderm explants, maternal deposition of dorsal determinants in cells close to the future wounding site is required for Nodal signaling activation and mesendoderm induction, much more than in embryos that retain Nodal signals emanating from the YSL (*Figure 4D,E*, *Figure 4—figure supplement 1E–H*, *Figure 4—figure supplement 2A–F*). To exclude that the effect of β-cat depletion on Nodal signaling is a mere secondary consequence of embryo ventralization (*Mizuno et al., 1999*; *Ober and Schulte-Merker, 1999*), we analyzed Nodal signaling in blastoderm explants prepared from embryos ventralized by overexpression of a constitutively active (CA) form of the BMP receptor Alk8 (*Payne et al., 2001*). CA-Alk8-overexpressing blastoderm explants did not display a strong reduction in the establishment of a Nodal

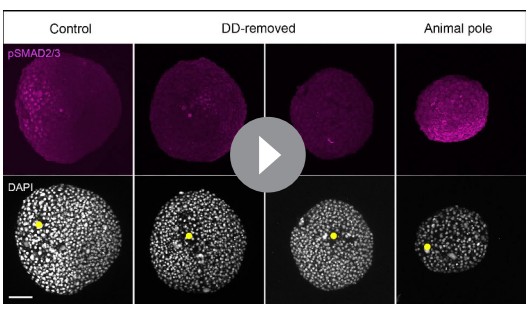

**Video 4.** Maternal pre-patterning is required for Nodal signaling in blastoderm explants. High-resolution fluorescence images of 50% epiboly control (left; n = 73, N = 14), dorsal determinants-removed (DD-removed, middle; n = 30, N = 7) and animal pole (right; n = 22, N = 6) explants stained for both pSMAD2/3 (pink) and DAPI (grey). A full stack projection (also shown in *Figure 4D*) for each condition is shown first, followed by the individual z stacks. The explants are shown as top views and the yellow circles denote the wounding site. Scale bar: 100 μm.
https://elifesciences.org/articles/55190#video4

signaling domain (*Figure 4—figure supplement 2G,H*), supporting the notion that β-cat does not affect Nodal signaling through its effect on dorsoventral embryo patterning.

To further challenge the notion that maternal pre-patterning of blastoderm explants is critical for mesendoderm induction and morphogenesis, we sought to prepare explants lacking all pre-patterning. To this end, we prepared explants from the animal pole (AP) of the blastoderm, which should be devoid of marginal cells and dorsal determinants (*Xu et al., 2014*). In these explants, no accumulation of β-cat was observed in the nuclei of cells close to the wounding site (*Figure 4—figure supplement 1C,D*). Moreover, AP-derived explants showed strongly reduced nuclear pSMAD2/3 accumulation, mesendoderm induction and tissue extension (*Figure 4D–H*, *Figure 4—figure supplement 1E–H*, *Video 4*). These findings therefore suggest that maternal pre-patterning by dorsal determinants is needed for proper mesendoderm induction and morphogenesis in blastoderm explants.

To determine whether positional information retained within the blastoderm and subsequently wounding site of blastoderm explants is critical for coordinated mesendoderm induction and patterning, we dissociated blastoderm explants obtained from high stage embryos into their constituent cells and allowed them to reaggregate in culture in the presence of serum-free medium (*Figure 4—figure supplement 3A*). These reaggregates are thus expected to consist of a random mixture of blastomeres, including some marginal cells that are pre-patterned to express Nodal signals and, thus, induce mesendoderm. We found that blastomere reaggregates underwent progressive compaction and eventually rounded up in culture (*Figure 4—figure supplement 3B*). Importantly, reaggregated cells were still capable of undergoing cell divisions (*Figure 4—figure supplement 3B*), as evidenced by their progressive reduction in cell diameter (*Figure 4—figure supplement 3C*), suggesting that blastomere reaggregates are viable under culture conditions. Interestingly, we found that even when cultured until stage-matched embryos had reached bud stage, blastomere reaggregates failed to form a clearly recognizable extension (*Figure 4—figure supplement 3D,E*). Consistent with this, we found that expression of the mesodermal marker *ntl* (*Schulte-Merker, 1995*; *Schulte-Merker et al., 1994*) was restricted to a few sparse cells or seemingly randomly scattered cells within the reaggregates (*Figure 4—figure supplement 3F–H*). Importantly, neither increasing the cell number within reaggregates nor changing the culture media conditions substantially impacted on reaggregate morphogenesis or its ability to undergo coordinated mesoderm specification (*Figure 4—figure supplement 3D–H*). Collectively, these findings suggest that maternal pre-patterning and positional information within the blastoderm and, subsequently wounding site of blastoderm explants, is required for coordinated mesoderm induction and morphogenesis.

Our analysis of blastoderm explants and reaggregates so far indicates that in isolation from extraembryonic signaling sources, the spatially localized pre-patterning within the blastoderm margin can still drive a large degree of mesendoderm patterning and morphogenesis. This seems to contradict previous claims from embryo analysis that Nodal signaling from the extraembryonic YSL is needed for these processes (*Carvalho and Heisenberg, 2010*; *Fan et al., 2007*; *Gagnon et al., 2018*; *Hong et al., 2011*; *van Boxtel et al., 2015*; *Veil et al., 2018*; *Xu et al., 2012*). To understand this apparent discrepancy, we directly compared mesendoderm induction in blastoderm explants with intact embryos where Nodal signal production within the YSL is blocked. To block Nodal expression within the YSL, we injected *sqt* and *cyc* MOs specifically into the YSL of high stage embryos. Consistent with previous observations (*Fan et al., 2007*), we found that injection of high amounts of both *sqt* (2 ng/embryo) and *cyc* (4 ng/embryo) MOs impaired induction of endoderm (*sox32*) and head mesoderm (*hgg* and *gsc*) (*Figure 5A*, *Figure 5—figure supplement 1A,B*), the two mesendoderm progenitor cell types whose induction is thought to require the highest dose of Nodal signaling (*Dougan et al., 2003*; *Gritsman et al., 2000*; *Schier et al., 1997*; *Thisse and Thisse, 1999*). The induction of notochord (*ntl*), paraxial (*papc*) and ventrolateral mesoderm (*myoD* and *tbx6*), in contrast, appeared normal in the morphant embryos (*Figure 5A*, *Figure 5—figure supplement 1A,B*). This also became apparent by 32 hpf, when most morphant embryos displayed severe cyclopia, indicative of defective induction of anterior mesendoderm structures, but largely normal trunk and tail structures (*Figure 5B*). Together, these observations suggest that - consistent with previous findings (*Fan et al., 2007*) - the YSL is needed for robust induction of endoderm and head mesoderm, and that this signaling source might be specifically required to achieve peak levels of Nodal signaling along the blastoderm margin.

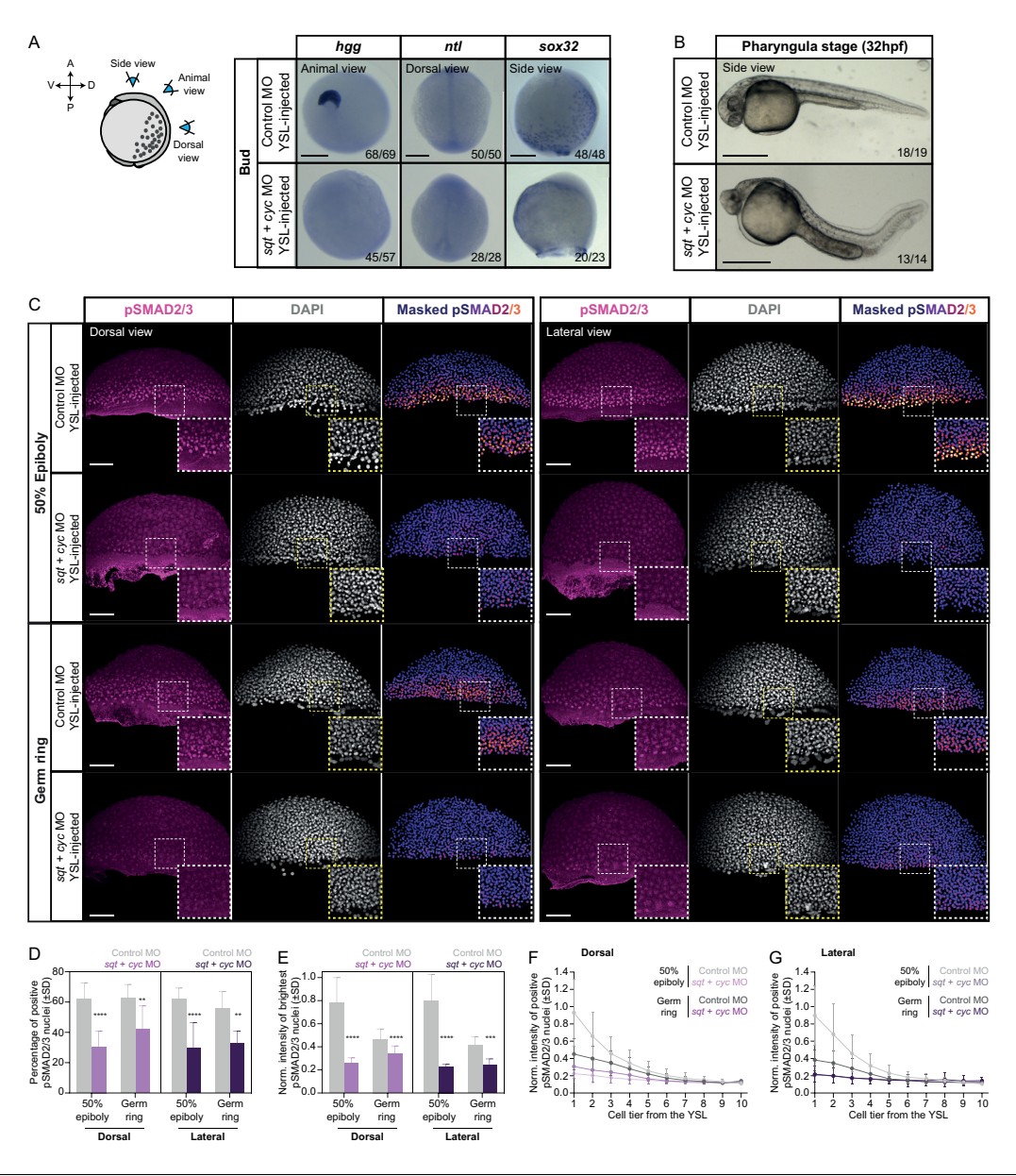

**Figure 5.** YSL-derived Nodal signals are required for inducing head mesoderm and endoderm in the intact embryo. (A) Expression of mesendoderm (*hgg* and *ntl*) and endoderm (*sox32*) marker genes, as determined by whole mount *in situ* hybridization in control MO (6 ng) or *sqt* (2 ng) + *cyc* (4 ng) MO YSL-injected embryos at bud stage. Schematic representation of the embryo views is shown on the left. The proportion of embryos with a phenotype similar to the images shown is indicated in the lower right corner (*hgg*: control, n = 69, N = 4, *sqt/cyc*, n = 57, N = 4; *ntl*: control, n = 50, N = 4, *sqt/cyc*, n = 28, N = 4 and *sox32*: control, n = 48, N = 3, *sqt/cyc*, n = 23, N = 3). (B) Bright-field single-plane images of pharyngula stage (32 hpf) control MO (6 ng) or *sqt* (2 ng) + *cyc* (4 ng) MO YSL-injected embryos. The proportion of embryos with a phenotype similar to the images shown is indicated in the lower right corner (n = 19, N = 2; n = 14, N = 2). (C) High-resolution fluorescence images of control MO (6 ng) or *sqt* (2 ng) + *cyc* (4 ng) MO YSL-injected embryos stained both for pSMAD2/3 (pink) and DAPI (grey) at 50% epiboly (dorsal domain: control, n = 10, N = 4; *sqt/cyc*, n = 7, N = 4; lateral domain: control, n = 9, N = 4; *sqt/cyc*, n = 7, N = 4) and germ ring (dorsal domain: control, n = 7, N = 4; *sqt/cyc*, n = 8, N = 4; lateral domain: control, n = 8, N = 4; *sqt/cyc*, n = 7, N = 4). Nuclear pSMAD2/3 is color-coded using a fire lookup table (highest intensities in yellow) and was masked based on the DAPI signal. Insets are zoom-in images of the highlighted regions (dashed boxes). (D) Percentage of pSMAD2/3 positive nuclei in control MO (6 ng) or *sqt* (2 ng) + *cyc* (4 ng) MO YSL-injected embryos at 50% epiboly (dorsal domain: control, n = 10, N = 4; *sqt/cyc*, n = 7, N = 4; lateral domain: control, n = 9, N = 4; *sqt/cyc*, n = 7, N = 4) and germ ring (dorsal domain: control, n = 7, N = 4; *sqt/cyc*, n = 8, N = 4; lateral domain: control, n = 8, N = 4; *sqt/cyc*, n = 7, N = 4). ****p<0.0001, **p=0.0093, **p=0.0023, respectively (ANOVA test). (E) Normalized intensity of the brightest pSMAD2/3 positive nuclei (for details see Materials and methods) in control MO (6 ng) or *sqt* (2 ng) + *cyc* (4 ng) MO YSL-injected embryos at 50% epiboly (dorsal domain: control, n = 10, N = 4; *sqt/cyc*, n = 7, N = 4; lateral domain: control, n = 9, N = 4; *sqt/cyc*, n = 7, N = 4) and germ ring (dorsal

*Figure 5 continued on next page*

*Figure 5 continued*

domain: control, n = 7, N = 4; *sqt/cyc*, n = 8, N = 4; lateral domain: control, n = 8, N = 4; *sqt/cyc*, n = 7, N = 4). ****p<0.0001, ***p=0.0010 (Kruskal-Wallis test). (F,G) Normalized intensity of the brightest pSMAD2/3 positive nuclei as a function of the distance to the YSL, expressed as cell tiers, in control MO (6 ng) or *sqt* (2 ng) + *cyc* (4 ng) MO YSL-injected embryos at 50% epiboly (for details see Materials and methods; dorsal domain: control, n = 10, N = 4; *sqt/cyc*, n = 7, N = 4; lateral domain: control, n = 9, N = 4; *sqt/cyc*, n = 7, N = 4) and germ ring (dorsal domain: control, n = 7, N = 4; *sqt/cyc*, n = 8, N = 4; lateral domain: control, n = 8, N = 4; *sqt/cyc*, n = 7, N = 4). (C-G) The position along the dorsal-ventral axis is indicated at the top. Scale bars: 200 µm (A), 500 µm (B), 100 µm (C).

The online version of this article includes the following source data and figure supplement(s) for figure 5:

**Source data 1.** YSL-derived Nodal signals are required for inducing head mesoderm in the intact embryo.
**Figure supplement 1.** Mesendoderm patterning in embryos with reduced YSL-derived Nodal signals.
**Figure supplement 1—source data 1.** Mesendoderm patterning in embryos with reduced YSL-derived Nodal signals.

To address this possibility, we analyzed the number and intensity of nuclear pSMAD2/3 in cells along the blastoderm margin of embryos injected with control (6 ng/embryo) or *sqt* (2 ng/embryo) and *cyc* (4 ng/embryo) MOs into the YSL. We found a strong and persistent reduction of both the number and intensity of nuclear pSMAD2/3 positive cells along the blastoderm margin in *sqt/cyc* morphants (*Figure 5C–G*, *Figure 5—figure supplement 1C*), indicating that YSL-derived Nodal signals are needed to enhance Nodal signaling in the overlying blastoderm. Next, we asked whether injecting lower amounts of *sqt/cyc* MOs into the YSL would result in a less pronounced reduction in endoderm and head mesoderm specification. YSL-specific injection of lower amounts of *sqt* (1 or 0.4 ng/embryo) and *cyc* (2 or 0.8 ng/embryo) MOs led to a dose-dependent reduction in both the number of morphant embryos expressing the ppl marker *hgg* and the size of the ppl, as determined by the expression area of *hgg*, in those embryos at bud stage (*Figure 6A–C*). Likewise, by 32 hpf, the severity of the cyclopia phenotype closely correlated with the amount of *sqt/cyc* MOs injected within the YSL (*Figure 5B*, *Figure 6—figure supplement 1A*). Endoderm induction, visualized by *sox32* expression (*Kikuchi et al., 2001*), was also strongly reduced in morphant embryos even with the lowest amount of YSL-injected *sqt/cyc* MOs (0.4 ng *sqt* + 0.8 ng *cyc* MO; *Figure 6—figure supplement 1B,C*). Consistent with such dose-dependent reduction in head mesoderm and endoderm specification, we found that Nodal signaling, assayed by pSMAD2/3 nuclear localization, was decreased to a lesser extent in embryos YSL-injected with low compared to high amounts of *sqt/cyc* MOs (2 ng *sqt* and 4 ng *cyc* MO) (*Figure 6D–G*, *Figure 6—figure supplement 1D*; compare to *Figure 5C–G*). Collectively, these findings indicate that robust induction of head mesoderm and endoderm structures in intact embryos critically relies on the right dosage of extraembryonic Nodal ligands.

Finally, to understand how the variable induction of head mesoderm in *sqt/cyc* MO morphants might relate to our observations in blastoderm explants, we performed a detailed analysis of the induction of these structures within the explants. Notably, we found that in explants the induction of head mesoderm and endoderm was highly variable (*Figure 1D*, *Figure 1—figure supplement 1G, H*), and both the size of the ppl and the number of *sox32*-positive endoderm cells were strongly reduced (*Figure 6C*, *Figure 1—figure supplement 1G,H,J*). This is consistent with our observations from the embryo analysis that Nodal signals from the extraembryonic YSL are needed for robust induction of these structures.

## Discussion

The distinct ability of embryonic or stem cell aggregates to autonomously generate *pattern* and *form* has challenged the classical view that maternal factors and/or extrinsic biases, such as extraembryonic tissues, are essential for proper embryo patterning and morphogenesis (*Mummery and Chuva de Sousa Lopes SM, 2013*; *Langdon and Mullins, 2011*; *Shahbazi et al., 2019*; *Simunovic and Brivanlou, 2017*; *Stern and Downs, 2012*; *Tam and Loebel, 2007*; *Turner et al., 2016*). By establishing a novel model system to culture zebrafish embryos in the absence of the yolk cell and its associated structures, we were able to show that blastoderm explants can recapitulate a remarkable degree of patterning and morphogenesis reminiscent of intact embryos. Similar observations were also recently reported in a preprint (*Trivedi et al., 2019*), supporting the notion that zebrafish embryonic tissues can autonomously organize into the structures they are primed to form.

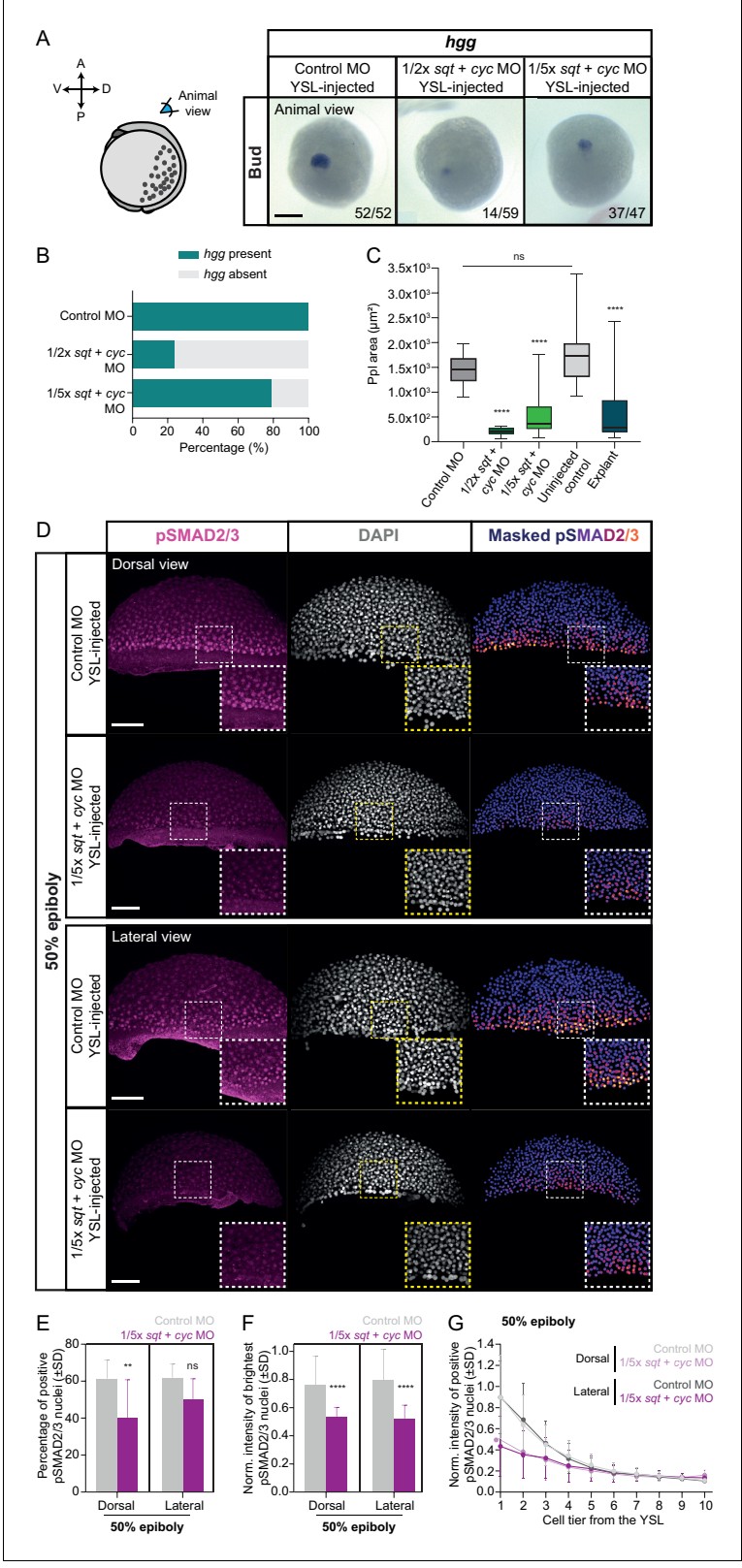

**Figure 6.** Reducing the dosage of YSL-derived Nodal signals results in variable induction of head mesoderm in intact embryos. (**A**) Expression of *hgg*, a head mesoderm marker gene, in bud stage control MO (6 ng) or varying dosages of *sqt + cyc* MO YSL-injected embryos (1/2x: 1 ng *sqt* + 2 ng *cyc* MO and 1/5x: 0.4 ng *sqt* + 0.8 ng *cyc* MO), as determined by whole mount *in situ* hybridization. All embryos are shown as an animal view (schematic

*Figure 6 continued on next page*

*Figure 6 continued*

representation in the left). The proportion of embryos with a phenotype similar to the images shown is indicated in the lower right corner (control MO: n = 52, N = 3; 1/2x *sqt/cyc*: n = 59, N = 3; 1/5x *sqt/cyc*: n = 47, N = 3). (**B**) Percentage of control MO (n = 52, N = 3) or *sqt + cyc* MO YSL-injected embryos (1/2x *sqt/cyc*: n = 59, N = 3; 1/5x *sqt/cyc*: n = 47, N = 3) showing expression of *hgg* as determined by whole mount *in situ* hybridization at bud stage. (**C**) Ppl area (based on *hgg* staining, as determined by whole mount *in situ* hybridization) in control MO (n = 52, N = 3) or *sqt + cyc* MO YSL-injected embryos (1/2x *sqt/cyc*: n = 14, N = 3; 1/5x *sqt/cyc*: n = 37, N = 3), uninjected wildtype embryos (n = 31, N = 4) or blastoderm explants, prepared from 256 c embryos (n = 20, N = 5). ****p<0.0001, ns, not significant. (Kruskal-Wallis test). (**D**) High-resolution fluorescence images of control MO (6 ng) or 1/5x *sqt* (0.4 ng) + *cyc* (0.8 ng) MO YSL-injected embryos stained both for pSMAD2/3 (pink) and DAPI (grey) at 50% epiboly (dorsal domain: control, n = 11, N = 5; 1/5x *sqt/cyc*, n = 8, N = 3; lateral domain: control, n = 9, N = 4; 1/5x *sqt/cyc*, n = 7, N = 3). Nuclear pSMAD2/3 is color-coded using a fire lookup table (highest intensities in yellow) and was masked based on the DAPI signal. Insets are zoom-in images of the highlighted regions (dashed boxes). (**E**) Percentage of pSMAD2/3 positive nuclei in control MO (6 ng) or 1/5x *sqt* (0.4 ng) + *cyc* (0.8 ng) MO YSL-injected embryos at 50% epiboly (dorsal domain: control, n = 11, N = 5; 1/5x *sqt/cyc*, n = 8, N = 3; lateral domain: control, n = 9, N = 4; 1/5x *sqt/cyc*, n = 7, N = 3). **p=0.0086, ns, not significant (Kruskal-Wallis test). (**F**) Normalized intensity of the brightest pSMAD2/3 nuclei (for details see Materials and methods) in control MO (6 ng) or 1/5x *sqt* (0.4 ng) + *cyc* (0.8 ng) MO YSL-injected embryos at 50% epiboly (dorsal domain: control, n = 11, N = 5; 1/5x *sqt/cyc*, n = 8, N = 3; lateral domain: control, n = 9, N = 4; 1/5x *sqt/cyc*, n = 7, N = 3). ****p<0.0001 (Kruskal-Wallis test). (**G**) Normalized intensity of pSMAD2/3 positive nuclei as a function of the distance to the YSL, expressed as cell tiers, in control MO (6 ng) or 1/5x *sqt* (0.4 ng) + *cyc* (0.8 ng) MO YSL-injected embryos at 50% epiboly (see Materials and methods for additional details; dorsal domain: control, n = 11, N = 5; 1/5x *sqt/cyc*, n = 8, N = 3; lateral domain: control, n = 9, N = 4; 1/5x *sqt/cyc*, n = 7, N = 3). (D-G) The position along the dorsal-ventral axis is indicated in the top right corner or at the bottom. Part of the control MO samples for (**A-G**) is also shown in *Figure 5A and C–G*. Scale bars: 200 μm (**A**), 100 μm (**D**).

The online version of this article includes the following source data and figure supplement(s) for figure 6:

**Source data 1.** Reducing the dosage of YSL-derived Nodal signals results in variable induction of head mesoderm in intact embryos.

**Figure supplement 1.** Mesendoderm patterning in embryos with varying dosages of YSL-derived Nodal signals.

**Figure supplement 1—source data 1.** Mesendoderm patterning in embryos with varying dosages of YSL-derived Nodal signals.

---

Moreover, our findings provide insight into the specific contributions of maternal pre-patterning and extraembryonic tissues for robust development.

Previous studies have provided evidence that maternal factors, initially deposited at the vegetal pole of the oocyte, are required to activate β-cat at the future dorsal side of the embryo and, consequently, initiate Nodal signaling therein (*Langdon and Mullins, 2011*; *Marlow, 2020*). Our data support this view and show that this maternal activation of Nodal signaling at the dorsal germ ring margin is required for proper mesoderm induction in blastoderm explants. In particular, our findings that removing dorsal and marginal pre-patterning in AP blastoderm explants or interfering with the spatial distribution of these pre-patterning cues in reaggregates is sufficient to suppress coordinated mesoderm induction and morphogenesis, clearly shows that maternal pre-patterning is indispensable for blastoderm explant organization and morphogenesis. This differs from recent observations in human and mouse embryonic stem cell (hESCs and mESCs, respectively) colonies, where embryoid bodies (EBs) are thought to form by spontaneous symmetry-breaking and self-organization (*Turner et al., 2016*). Possibly, this difference in self-organizing capacity is due to blastoderm cells displaying reduced stemness when compared to ES cells, a plausible assumption given that blastoderm explants were taken from intact embryos that already underwent some degree of embryo polarization and cell fate restriction (*Langdon and Mullins, 2011*; *Marlow, 2020*). Furthermore, we cultured our blastoderm explants in media without serum or other growth factors in order to monitor blastoderm maturation in the absence of any extrinsic signals. In contrast, stem cell colonies or, most recently, also zebrafish explants were cultured in media supplemented with growth factors and/or serum, which might be required for self-organization to emerge *in vitro* (*Baillie-Johnson et al., 2015*; *Beccari et al., 2018*; *ten Berge et al., 2008*; *Turner et al., 2017*; *Turner et al., 2014*; *Trivedi et al., 2019*; *van den Brink et al., 2014*; *Warmflash et al., 2014*).

In zebrafish embryos, extraembryonic Nodal signals from the YSL are thought to be required for mesendoderm induction along the dorsal-ventral axis (*Carvalho and Heisenberg, 2010*; *Fan et al., 2007*; *Gagnon et al., 2018*; *Hong et al., 2011*; *Schier and Talbot, 2005*; *van Boxtel et al., 2015*; *Veil et al., 2018*; *Xu et al., 2012*). Our finding that blastoderm explants can not only induce the three germ layers, but also display a seemingly complete patterning of the mesendoderm anlage, seems to argue against such critical function for YSL-derived Nodal signals in mesendoderm patterning. However, closer inspection of the role of the YSL for mesendoderm patterning revealed some striking similarities between embryos with reduced YSL-derived Nodal signals and blastoderm explants: in both cases, induction of head mesoderm and endoderm, two tissues whose induction requires high doses of Nodal signaling (*Dougan et al., 2003*; *Gritsman et al., 2000*; *Schier et al., 1997*; *Thisse and Thisse, 1999*), is rather variable. This points at the intriguing possibility that Nodal signals from the extraembryonic YSL effectively boost Nodal signaling within the blastoderm margin, needed for robust induction of these two tissues. Yet, mesendoderm patterning defects in embryos with reduced Nodal signals from the YSL are generally more pronounced than in blastoderm explants, suggesting that the YSL is more important for robust induction of head mesoderm and endoderm in the intact embryo compared to explants. One possibility for such embryo-specific requirement of the YSL might be to induce sufficiently high doses of Nodal signaling along the entire blastoderm margin, a requirement less important in explants where the blastoderm margin has collapsed into a single region surrounding the wounding site.

While blastoderm explants were able to recapitulate a remarkable degree of embryo patterning and morphogenesis, there are still several differences between explants and embryos: foremost, mesendoderm progenitors do not undergo internalization in explants, but instead form an 'exogastrula', similar to previous observations in mouse gastruloids (*Baillie-Johnson et al., 2015*; *Beccari et al., 2018*; *Turner et al., 2017*; *Turner et al., 2014*; *van den Brink et al., 2014*). One possible explanation for this could be that the specific geometry of the embryo is critical for mesendoderm internalization. Alternatively, extraembryonic structures might form a permissive substrate for mesendoderm internalization. Notably, while mesendoderm progenitors fail to internalize in explants, they still form an extension, which - similar to body axis elongation in embryos - depends on Wnt/PCP-signaling (*Čapek et al., 2019*; *Gray et al., 2011*; *Solnica-Krezel and Sepich, 2012*; *Tada and Heisenberg, 2012*). This suggests that the cellular rearrangements underlying body axis extension in explants and body axis elongation in embryos might be conserved. Moreover, our and previous observations support that cell proliferation does not seem to play a major role in either of these processes (*Liu et al., 2017*; *Quesada-Hernández et al., 2010*; *Zhang et al., 2008*). Yet, whether volumetric growth by cell proliferation might function in later stages of explant maturation, similar to its role, for instance, in tail extension in chick and mouse embryos (*Mongera et al., 2019*), remains to be determined.

Collectively, our findings suggest that zebrafish blastoderm explants undergo what has been recently termed 'genetically encoded self-assembly', where heterogeneities among cells endowed with a similar genetically encoded cell or developmental program drive the organization of those cells, via local interactions, into the structures they were initially primed to form (*Turner et al., 2016*; *Xavier da Silveira Dos Santos and Liberali, 2019*). This interpretation implies that both intrinsic heterogeneities (e.g. maternal factors) and extraembryonic signals might not only be required to render embryonic development predictable and robust, but also to drive the emergence of embryo organization. Future work will have to address to what extent stem cell colonies/organoids indeed undergo bona fide self-organization, and whether some of their organization might in fact be due to genetically encoded self-assembly.

## Materials and methods

**Key resources table**

| Reagent type (species) or resource | Designation | Source or reference | Identifiers | Additional information |
| --- | --- | --- | --- | --- |
| Strain, strain background (*D. rerio*) | Zebrafish: wildtype ABxTL | MPI-CBG dresden | | |

*Continued on next page*

*Continued*

| Reagent type (species) or resource | Designation | Source or reference | Identifiers | Additional information |
|---|---|---|---|---|
| Strain, strain background (*D. rerio*) | Zebrafish: Tg(*gsc::EGFP-CAAX*) | (*Smutny et al., 2017*) | ZFINID:ZDB-ALT-170811–2 | |
| Strain, strain background (*D. rerio*) | Zebrafish: Tg(*sebox::EGFP*) | (*Ruprecht et al., 2015*) | ZFINID:ZDB-ALT-150727–1 | |
| Strain, strain background (*D. rerio*) | Zebrafish: Tg(*krt4::EGFP-CAAX*) | (*Krens et al., 2011*) | ZFINID:ZDB-ALT-111207–5 | |
| Strain, strain background (*D. rerio*) | Zebrafish: Tg(*actb2::lyntdtomato;actb2::H2B-EGFP*) | (*Stegmaier et al., 2016*) | ZFINID:ZDB-ALT-161017–9 | |
| Strain, strain background (*D. rerio*) | Zebrafish: MZoep | (*Gritsman et al., 1999*) | ZFINID:ZDB-ALT-980203–1256 | |
| Strain, strain background (*D. rerio*) | Zebrafish: MZfz7a/b | (*Quesada-Hernández et al., 2010*) | ZFINID:ZDB-FISH-150901–19586 | |
| Strain, strain background (*D. rerio*) | Zebrafish: MZfz7a/b; Tg(*gsc::EGFP-CAAX*) | (*Čapek et al., 2019*) | | |
| Strain, strain background (*D. rerio*) | Zebrafish: MZwnt11 | (*Heisenberg et al., 2000*) | ZFINID:ZDB-ALT-980203–1302 | |
| Strain, strain background (*D. rerio*) | Zebrafish: MZwnt5b | (*Kilian et al., 2003*) | ZFINID:ZDB-ALT-980203–1630 | |
| Recombinant DNA reagent | pCS2+-Membrane-GFP plasmid for mRNA synthesis | (*Kimmel and Meyer, 2010*) | | 60–100 pg (1 c injection) |
| Recombinant DNA reagent | pCS2-Membrane-RFP plasmid for mRNA synthesis | (*Iioka et al., 2004*) | | 80 pg (1 c injection) |
| Recombinant DNA reagent | pCS2+-H2B-eGFP plasmid for mRNA synthesis | (*Keller et al., 2008*) | | 50 pg (1 c or YSL injection) |
| Recombinant DNA reagent | pCS2-H2A-mCherry plasmid for mRNA synthesis | (*Arboleda-Estudillo et al., 2010*) | | 6.5 pg (128 c injection) 50 pg (1 c injection) |
| Recombinant DNA reagent | pCS2+-CA-Alk8 plasmid for mRNA synthesis | (*Payne et al., 2001*) | | 30 pg (1 c injection) |
| Sequence-based reagent | Human β-Globin Morpholino: 5′-CCTCTTACCTCAGTTACAATTTATA-3′ | Gene Tools | | 6 ng (YSL injection) |
| Sequence-based reagent | *sqt* Morpholino: 5′-ATGTCAAATCAAGGTAATAATCCAC-3′ | Gene Tools | ZFINID:ZDB-MRPHLNO-060809–1 | 0.4–2 ng (YSL injection) |
| Sequence-based reagent | *cyc* Morpholino: 5′-GCGACTCCGAGCGTGTGCATGATG-3′ | Gene Tools | ZFINID:ZDB-MRPHLNO-060930–4 | 0.8–4 ng (YSL injection) |
| Antibody | Anti-Phospho-Smad2 (Ser465/467)/Smad3 (Ser423/425) (clone D27F4) | Cell Signaling | Cat# 8828, RRID:AB_2631089 | 1:1000 (WMIF) |
| Antibody | Anti-β-Catenin (clone 15B8) | Sigma-Aldrich | Cat# C7207, RRID:AB_476865 | 1:500 (WMIF) |
| Antibody | Anti-Phospho-Smad1/5 (Ser463/465) (clone 41D10) | Cell Signaling | Cat#: 9516, RRID:AB_491015 | 1:100 (WMIF) |
| Antibody | Alexa Fluor 546 Goat Anti-rabbit IgG (H + L) | Thermo Fisher Scientific | Cat#: A-11010; RRID:AB_2534077 | 1:500 (WMIF) |
| Antibody | Alexa Fluor 488 Goat Anti-mouse IgG (H + L) | Thermo Fisher Scientific | Cat#: A-11001; RRID:AB_2534069 | 1:500 (WMIF) |
| Antibody | Alexa Fluor 647 Goat Anti-mouse IgG (H + L) | Thermo Fisher Scientific | Cat# A-21235; RRID:AB_2535804 | 1:500 (WMIF) |

*Continued on next page*

*Continued*

| Reagent type (species) or resource | Designation | Source or reference | Identifiers | Additional information |
|---|---|---|---|---|
| Antibody | Anti-Digoxigenin-AP Fab fragments | Roche | Cat#11093274910 RRID:AB_2734716 | 1:1000 (WMISH) |
| Antibody | Anti-Fluorescein-AP Fab fragments | Roche | Cat#11426338910 RRID :AB_2734723 | 1:500 (WMISH) |
| Chemical compound, drug | DMSO | Sigma-Aldrich | Cat#D8418 | 50 µM |
| Chemical compound, drug | SB-505124 | Sigma-Aldrich | Cat# S4696 | 50 µM |
| Chemical compound, drug | Dextran Alexa Fluor 647 | Invitrogen | Cat#D22914 | 2 ng (Sphere stage injection) |
| Chemical compound, drug | Dextran Cascade Blue | Invitrogen | Cat# D1976 | 2 ng (Sphere stage injection) |
| Chemical compound, drug | Hydroxyurea | Sigma-Aldrich | Cat#H8627 | 60 mM |
| Chemical compound, drug | Aphidicolin | Sigma-Aldrich | Cat#A0781 | 300 µM |
| Chemical compound, drug | DAPI | Invitrogen | Cat#D1306 | 1 µg/mL (WMIF) |
| Chemical compound, drug | Methylcellulose | Sigma-Aldrich | Cat#M0387 | 0.3% |
| Chemical compound, drug | Digoxigenin (DIG)-modified nucleotides | Sigma-Aldrich | Cat#11277073910 | |
| Chemical compound, drug | Fluorescein-labeled nucleotides | Sigma-Aldrich | Cat#11685619910 | |
| Chemical compound, drug | SIGMAFAST Fast Red TR/Naphthol AS-MX Tablets | Sigma-Aldrich | Cat#F4648 | |
| Cell culture reagent | L15 | Sigma-Aldrich | Cat# L4386 | |
| Software, Algorithm | Imaris | Bitplane | https://imaris.oxinst.com/packages | |
| Software, Algorithm | Fiji | (*Schindelin et al., 2012*) | https://fiji.sc/ | |
| Software, Algorithm | GraphPad Prism | GraphPad Software | https://www.graphpad.com/scientific-software/prism/ | |
| Software, Algorithm | MATLAB | MATLAB Software | https://www.mathworks.com/products/ matlab.html | |
| Software, Algorithm | pSMAD2/3 and β-catenin analysis (custom-made script) | This paper | *Source code 1* | |
| Software, Algorithm | Excel | Microsoft | https://products.office.com/en-us/?rtc=1 | |

## Fish lines and husbandry

Zebrafish (*Danio rerio*) handling was performed as described (*Westerfield, 2000*). The mutant and transgenic lines used in this study are detailed in the key resources table. Embryos were raised at 25–31°C in Danieau's solution or E3 medium and staged according to *Kimmel et al. (1995)*.

## Blastoderm explant preparation

Embryos for blastoderm explant preparation were manually dechorionated with watchmaker forceps. Explants were then prepared at 256 cell stage, unless stated otherwise, by removing the entire blastoderm from the yolk cell using forceps, and then washed by gentle pipetting to remove residual yolk granules (schematic representation in *Figure 1A*). Animal pole explants were prepared by

removing only the animal portion of the blastoderm at 256 cell stage using a hair knife instead of forceps. All explants were cultured in Ringer's solution (116 mM NaCl, 2.9 mM KCl, 1.8 m M CaCl$_2$, 5.0 mM HEPES, pH 7.2) or Danieau's medium (58 mM NaCl, 0.7 mM KCl, 0.4 mM MgSO$_4$, 0.6 mM Ca(NO$_3$)$_2$, 5 mM HEPES, pH 7.6). Approximately 1 hour (hr) after explant preparation, explants with delayed cell cleavages, as assessed by increased cell size, were removed from the dish. The remaining explants were then staged based on sibling embryos from the same egg lay. To analyze explant morphology, bright-field side view images of blastoderm explants were acquired using a stereomicroscope (Olympus SZX 12) equipped with a QImaging Micropublisher 5.0 camera.

### Embryo microinjections

mRNAs were synthesized using the mMessage mMachine Kit (Ambion). Injections at 1 cell stage were carried out as described (*Westerfield, 2000*) and the plasmids used for mRNA production are detailed in the key resources table. Labelling the interstitial fluid was performed by injecting 2 ng of Dextran Alexa Fluor 647, or Dextran Cascade Blue, between the deep cells of blastoderm explants corresponding to high stage or sphere stage embryos, as described in *Krens et al. (2017)*. For time-lapse imaging of the interstitial fluid, embryos were also injected at 1 cell stage with 80 pg of Membrane-GFP. For high-resolution live imaging of gastrulating blastoderm explants, Tg(*gsc::EGFP-CAAX*) expressing embryos were injected at 1 cell stage with 50 pg of H2A-chFP to label the cell nuclei. For the clone dispersal analysis, the plasma membrane of all cells was labelled by injecting 80 pg of Membrane-RFP mRNA at 1 cell stage. 6.5 pg of H2A-chFP mRNA was subsequently injected into a single marginal blastomere at 128 cell stage (schematic representation in *Figure 2—figure supplement 2A*). To constitutively activate BMP signaling, 30 pg of CA-Alk8 mRNA were injected together with 60 pg of Membrane-GFP mRNA at 1 cell stage.

YSL injections were performed right after its formation, between 1 k and high stage. As a control for homogeneous YSL injections, 50 pg of H2B-EGFP mRNA and 0.2% Phenol Red were co-injected into the YSL. To block YSL-derived Nodal signals, 0.4–2 ng of *sqt* MO (*sqt* MO: 5′-ATGTCAAATCAAGGTAATAATCCAC-3′ [*Feldman and Stemple, 2001*]) were co-injected with 0.8–4 ng of *cyc* MO (*cyc* MO: 5′-GCGACTCCGAGCGTGTGCATGATG-3′ [*Karlen and Rebagliati, 2001*]) into the YSL. As a control for MO injections, 6 ng of a standard negative control MO (human β-globin MO: 5′-CCTCTTACCTCAGTTACAATTTATA-3′) were also injected.

### Live imaging of blastoderm explants

Bright-field time-lapse imaging of explants was performed using a Zeiss LSM 800 upright microscope equipped with a Zeiss Plan-Apochromat 20x/1.0 water immersion objective starting when stage-matched control embryos had reached sphere-early dome stage. High-resolution live imaging of developing blastoderm explants was performed using a LaVision Trim 2-photon microscope equipped with a Zeiss Plan-Apochromat 16x/0.8 water immersion objective and a Ti:Sa laser (Chameleon, Coherent) set to 830 nm and an OPO laser set at 1100 nm. In this case, imaging was started when stage-matched control embryos had reached 50% epiboly-germ ring stage. In both cases, blastoderm explants were oriented in a side view, and the acquired images were then processed using Fiji and/or Imaris.

### Analysis of fluid accumulation, YSL formation and EVL differentiation

Imaging of the explant interstitial fluid at bud stage was performed using a LaVision Trim 2-photon microscope equipped with a Zeiss Plan-Apochromat 20x/1.0 water immersion objective and a Ti:Sa laser (Chameleon, Coherent) set to 800 nm and an OPO laser set at 1100 nm, or a Zeiss LSM 880 upright confocal microscope equipped with a Zeiss Plan-Apochromat 20x/1.0 water immersion objective. Time-lapse imaging of the explant interstitial fluid was performed using a Zeiss LSM 800 upright microscope equipped with a Zeiss Plan-Apochromat 20x/1.0 water immersion objective, starting when stage-matched control embryos had reached high-oblong stage. In both cases, blastoderm explants were oriented in a side view, and the acquired images were then processed using Fiji and/or Imaris.

Imaging of YSL-like features in blastoderm explants was performed when corresponding stage-matched control embryos reached 50% epiboly-germ ring stage using a LaVision Trim 2-photon

microscope as described above. Explants were oriented with the wounding site facing the objective (top view), and the acquired images were then processed using Fiji.

To analyze EVL differentiation in blastoderm explants, both animal-pole oriented embryos and explants were imaged in parallel on a Zeiss LSM 880 upright confocal microscope equipped with a Zeiss Plan-Apochromat 20x/1.0 water immersion objective. To assess whether the entire explant was covered by EVL, explants and embryos were then fixed at bud stage in 4% paraformaldehyde (PFA) at 4℃ overnight, washed 5x with PBS and then imaged on a Zeiss LSM 880 upright confocal microscope equipped with a Zeiss Plan-Apochromat 20x/1.0 water immersion objective. The acquired images for both embryos (side view) and explants (side view) were then processed using Fiji and/or Imaris (Bitplane).

## Sample preparation for live imaging

Dechorionated embryos were mounted in 2% agarose molds on petri dishes and immobilized in 0.7% low melting point agarose. Explants were mounted in smaller agarose molds (800 × 800 μm from Microtissues) and cultured in Ringer's or Danieau's solution.

## Whole mount *in situ* hybridization (WMISH)

Embryos were fixed with 4% PFA overnight at 4℃. Antisense RNA probes were synthesized using SP6, T7 or T3 RNA polymerase from mMessage mMachine kits (ThermoFisher, AM1344) with Roche digoxigenin (DIG)-modified nucleotides, or fluorescein (FITC)-labeled nucleotides, from partial cDNA sequences. WMISHs were performed as previously described (*Thisse and Thisse, 2008*) with the following modifications: (i) proteinase K treatment was performed only for *sox32 in situ* hybridization and for double WMISHs; (ii) 5% dextran sulfate was added to the hybridization solution for *sox32*, *mxtx2*, *tbx16* and *bmp2b* probes and all double WMISHs; and, (iii) 0.05x SSC was used for more stringent washes for *sqt in situ* hybridizations. Single WMISHs were performed with DIG-labeled antisense RNA probes, while for double WMISHs, a mixture of DIG- and FITC-labeled RNA probes were added simultaneously during the hybridization step.

For double WMISHs, hybridization and visualization of DIG-labelled probes with the chromogenic NBT/BCIP substrate were performed as previously described (*Thisse and Thisse, 2008*). When the signal was sufficiently intense, the staining solution was replaced with a solution of 100 mM glycine-HCl pH2.2 for 10 min at room temperature as described in *Denker et al. (2008)*. The samples were then washed several times with fresh PBS + 0.1% (w/v) Tween (PBST) and blocked in blocking buffer (1 × PBST, 2% sheep serum (v/v), 2 mg/ml BSA) for 2 hr at room temperature and incubated overnight at 4℃ with Anti-Fluorescein-AP Fab fragments (1:500) in fresh blocking buffer. Afterwards, the samples were washed 6x for 10–20 min at room temperature in PBT and 3x in 0.1 M Tris-HCl pH8.2 + 0.1% (w/v) Triton. The staining solution for red staining was prepared using SIGMAFAST Fast Red TR/Naphthol AS-MX Tablets according to the manufacturer's protocol and added to the samples. After staining, the samples were cleared for 10 min in 100% EtOH and washed several times in PBT. Blastoderm explants and corresponding stage-matched embryos were always processed in the same tube, and the staining reaction was therefore stopped simultaneously. All WMISHs were imaged on a stereo-microscope (Olympus SZX 12) equipped with a QImaging Micropublisher 5.0 camera.

## Whole mount immunofluorescence (WMIF)

α-pSMAD2/3 and α-β-catenin whole mount immunofluorescence was performed as described previously (*van Boxtel et al., 2015*). In short, dechorionated embryos and explants were fixed in 4% PFA (in PBS) overnight at 4℃, washed in PBS and then directly transferred into 100% MeOH and stored at −20℃ for, at least, 2 hr. The samples were then washed 5x in PBSTr (PBS + 1% (w/v) Triton X-100), blocked 2x in blocking solution (PBSTr + 10% goat serum + 1% DMSO) for 2 hr and then incubated overnight at 4℃ with α-pSMAD2/3 antibody (1:1000) and/or α-β-catenin antibody (1:500) diluted in blocking solution. Afterwards, the samples were washed 5x in PBSTr for 10 min, washed 3x in PBS + 0.1% (w/v) Triton for 1 hr and then incubated overnight at 4℃ in secondary antibody (1:500) diluted in blocking solution. For labelling cell nuclei, the samples were incubated in DAPI (1:1000) diluted in blocking solution for 18 min at room temperature. After secondary antibody incubation, the samples were washed 2x for 5 min in PBSTr and 5x for 10–20 min in PBS + 0.1% Triton.

α-pSMAD1/5 whole mount immunofluorescence was performed as described previously (*Pomreinke et al., 2017*) with the following modifications: (i) for re-hydration, embryos and explants were directly transferred to PBST; (ii) Anti-Phospho-Smad1/5 (Ser463/465) (1:100) was used to detect pSmad1/5; (iii) secondary antibody incubation (1:500) was performed overnight at 4°C; and (iv) labelling of cell nuclei with DAPI was performed as described above. Explants and corresponding stage-matched control embryos were always processed in the same tube.

Immunostained embryos and blastoderm explants were imaged using an upright confocal microscope (Zeiss LSM 880 or a Zeiss LSM 800 upright microscope equipped with a Zeiss Plan-Apochromat 20x/1.0 water immersion objective) and mounted in 2% agarose molds and subsequently covered with PBS. Embryos and explants were oriented using the Tg(*gsc::EGFP-CAAX*) line to identify the dorsal side (embryos) or wounding site (explants). Imaging conditions were kept similar between different samples and replicates of the same experiment.

## Nodal inhibitor treatment

Blastoderm explants together with stage-matched control embryos were incubated in 0.1% DMSO or in 50 µM of the reversible Nodal inhibitor SB-505124 (*DaCosta Byfield et al., 2004*; *Fan et al., 2007*; *Hagos and Dougan, 2007*; *Hagos et al., 2007*; *Rogers et al., 2017*; *van Boxtel et al., 2015*; *Vogt et al., 2011*). For inhibitor washout experiments, embryos and explants were washed 5x in Danieau's solution for 5 min.

## Cell division inhibition (HUA treatment)

To block cell division events during explant extension, blastoderm explants were incubated in 0.1% DMSO or a cocktail of 60 mM Hydroxyurea and 300 µM Aphidicolin (HUA) from shield to bud stage, as determined by stage-matched control embryos. To assess explant morphology, bright-field side view images of bud stage blastoderm explants treated with DMSO or HUA were then acquired using a stereo-microscope (Olympus SZX 12) equipped with a QImaging Micropublisher 5.0 camera.

## Removal of dorsal determinants

Removal of dorsal determinants was performed as described (*Mizuno et al., 1999*; *Ober and Schulte-Merker, 1999*). In short, embryos were dechorionated immediately after egg laying in Danieau's solution and the vegetal-most part of the yolk cell was removed using a hair knife. The whole procedure was carried out within 15 min after the first eggs were laid. Manipulated embryos were then allowed to heal for 10 min before being transferred to fresh Danieau's medium. Dorsal determinants removed embryos were used as staging controls for the corresponding explants.

## Assessment of mesendoderm induction

For assessing changes in mesoderm induction upon Nodal inhibitor treatment, after dorsal determinants removal and in animal pole explants, corresponding explants were prepared from Tg(*sebox:: EGFP*) transgenic embryos to mark mesendoderm progenitors. At bud stage, these explants together with control explants were fixed in 4% PFA at 4°C overnight and washed 5x in PBS. The samples were then imaged on a Zeiss LSM 800 Upright microscope equipped with a Zeiss Plan-Apochromat 20x/1.0 water immersion objective. Imaging conditions were kept similar between all samples and replicates.

## Blastomere reaggregate preparation

For dissociation, manually dechorionated embryos were transferred to 1 ml of dissociation buffer (calcium-free Ringer's + 5 mM EGTA for reaggregates cultured in Ringer's solution and L15 + 5 mM EGTA for reaggregates cultured in L15) at high stage. The embryos were then mechanically dissociated by pipetting and washed 3x with the corresponding culture medium, followed by centrifugation at 200 g for 3 min to remove residual yolk proteins and EGTA. The dissociated cells were then seeded in agarose microwells cast from PDMS molds (Microtissues, diameter 800 µm, depth 800 µm) at a density of one or three dissociated embryo(s)/microwell. After a settlement period of around 30 min, the petri dish with dissociated cells was filled with the respective culture medium (schematic representation in *Figure 4—figure supplement 3A*). Blastomere reaggregates were

analyzed for their morphology and fixed for *in situ* hybridization, when control embryos had reached bud stage.

To assess whether cells keep dividing in culture - as a proxy for their health status and continued development - small reaggregates (one embryo/microwell), prepared from a mix of embryos with labelled nuclei and cell membranes (1 cell stage injection of 80–100 pg Membrane-GFP and 50 pg of H2A-chFP) and unlabeled embryos, were cultured in Ringer's solution. These reaggregates were then imaged at a Zeiss LSM 880 upright confocal microscope equipped with a Zeiss Plan-Apochromat 20x/1.0 water immersion objective shortly after reaggregation. The size of 15–30 randomly selected cells on the inside of the reaggregate was measured every 32 min using the Line tool in Fiji.

## Analysis of explant morphology

To quantify the morphology of blastoderm explants, bright-field side view images of explants were acquired as described above. Explants were considered as extended when a clear indentation was observed between the round 'back' and extended 'tip' of the explant. Explant circularity was then determined by manually outlining the shape of the explant using the Circularity plugin from Fiji. All explants, independently of whether they formed a clearly recognizable extension or not, were pooled for the circularity analysis.

To quantify the relative length of the explant extension, the extension length starting where a clear change in curvature was detectable was measured using the Segmented Line tool from Fiji and then normalized to the total explant length. Only explants with a recognizable extension were included in this analysis. Note that in this analysis, we only included blastoderm explants where the extension could be clearly delineated (except for *Figure 2—figure supplement 3D* and *Figure 4— figure supplement 1H*, where not-extended explants were included and their normalized extension length was considered to be 0). To ensure that potential defects in axis extension were indeed due to the different pharmacological and genetic perturbations and not embryo batch variability, only experiments in which at least 70% of control explants showed clearly recognizable extensions were considered for further analysis.

To analyze the change in area during explant extension, side view images of explants at the indicated time points were manually outlined using the freehand tool (schematic representation in *Figure 3—figure supplement 1A*). The width of the explants was measured at the point where the explant was thickest using the Line tool from Fiji (schematic representation in *Figure 3—figure supplement 1B*). The length of the extension over time was measured as described above (schematic representation in *Figure 3—figure supplement 1C*).

## Analysis of explant cavity localization

To quantify the average location of the explant cavity, bright-field side view images of explants were acquired as described above. The segmented line tool in Fiji was used to determine the distances from the tip of the extension to the closest edge of the cavity, and from the opposite edge of the cavity to the other end of the explant. The measured distances were then normalized to the total explant length, as schematically represented in *Figure 1—figure supplement 1B*. If an explant formed more than one cavity, the location of the biggest lumen was used for this quantification.

## Analysis of cell size

Cell size in blastoderm explants and stage-matched control embryos was measured using the Line tool in Fiji. For this, explants and embryos were fixed at various developmental stages (high stage, sphere, 30% epiboly, 50% epiboly, germ ring and shield stage) and immunostained for β-catenin to outline cell-cell contacts. Approximately 30 randomly selected deep cells from both explants and embryos were measured for this analysis.

## Analysis of gene expression domains

For this analysis, only blastoderm explants expressing the gene of interest, as determined by *in situ* hybridization, and where the back-tip axis could be clearly delineated, were analyzed. To quantify the location of a gene expression domain along the back-tip axis of the explant, the distances from the tip of the extension to the start and end of the expression domain were measured and normalized by the total explant length using the Segmented Line tool in Fiji. The normalized length of the

explant was then binned in 10 windows and the presence (1) or absence (0) of expression within a certain bin was plotted in a binary fashion and, ultimately, averaged across explants and experimental replicates. For *six3*-positive explants, which in some cases exhibited more than one expression domain, the measurement was performed for each expression domain separately. For *hgg*-positive explants, only samples with a single ppl were considered for this analysis.

To compare the size of the *hgg* expression domains in different experimental conditions, the boundaries of the expression domain, as determined by *in situ* hybridization, was manually outlined and measured in Fiji. Animal or side views were used respectively for intact embryos and blastoderm explants, and only samples with a single ppl were considered for this analysis.

## Analysis of EVL differentiation

Analysis of early EVL differentiation was performed using Tg(*krt4::EGFP-CAAX*) embryos, expressing EGFP in differentiating EVL cells. Analysis of EGFP expression in surface cells of both stage-matched embryos and explants , was used as a readout for EVL differentiation and this analysis was started shortly after the first visible accumulation of EGFP in the intact embryo. For quantifying EGFP accumulation, the same number of z planes were SUM projected in explants and embryos using Fiji. A ROI of 400.90 $\mu m^2$ was drawn in the center of the samples, and EGFP mean intensity was measured every 15 min. For background subtraction, an ROI of 24.91 × 24.91 $\mu m^2$ was analyzed outside of the sample.

## Analysis of pSMAD2/3 nuclear accumulation

For a quantitative analysis of pSMAD2/3 nuclear accumulation, randomly selected embryos and stage-matched blastoderm explants exhibiting pSMAD2/3 positive nuclei were used. The Imaris Spot detection plugin was used to automatically determine the 3D coordinates of all nuclei based on the DAPI signal. Nuclei belonging to the EVL, undergoing cell division or displayed inhomogeneous and/or barely visible DAPI signal, were manually excluded from the analysis. Moreover, only nuclei within a Z-volume of 100 µm were considered for this analysis in order to avoid depth-related intensity changes. The mean fluorescence intensities of pSMAD2/3 and DAPI as well as the corresponding nuclei 3D coordinates were extracted using the Spot Detection plugin. As a reference point to calculate nuclei distribution within the explants or embryos, the wounding site of the explant, detectable by increased and irregular junctional staining of β-catenin and/or comparatively lower and more irregular nuclei density, or the YSL nuclei in the intact embryo were manually determined using the same plugin and, subsequently, extracted. To determine the distance of each nucleus to the wounding site or YSL, a custom MATLAB script (*Source code 1*) was used to project all nuclei into a 2D plane along a manually defined axis. For embryos, all nuclei were projected along their Z axis. For explants, an additional reference point was manually provided and used to re-position the wounding site in the center of the image. The geometric distance of all nuclei to the closest YSL/wounding site spot in this 2D plane was then automatically calculated using the aforementioned script.

To determine the number of pSMAD2/3 positive nuclei in blastoderm explants and embryos, the following criteria were used: (i) only nuclei within 150 µm from the reference point (YSL nuclei for embryos, wounding site for explants) were considered in order to avoid imaging artifacts often arising at the sample edges; (ii) following background subtraction, only nuclei with positive nuclear pSMAD2/3 mean intensity were considered for further analysis; (iii) nuclear pSMAD2/3 mean intensity was normalized using DAPI mean intensity to correct depth-related artifacts; and (iv) only nuclei with a normalized pSMAD2/3 to DAPI ratio above 0.1 were considered to be positive. For background subtraction, the mean nuclear pSMAD2/3 intensity between 120–150 µm of the YSL or wounding site was used.

The brightest pSMAD2/3 positive nuclei were defined as the top 30% brightest normalized pSMAD2/3 to DAPI ratio of all positive nuclei, which typically correspond to the first 1–2 cell rows in the intact embryo. The brightest pSMAD2/3 positive nuclei in stage-matched explants and embryos were used for intensity comparison and for calculating the distance to the YSL or wounding site. To plot this distance as cell tiers, we considered the average cell size measured at 50% epiboly, germ ring and shield stage, as described above (see *Figure 1—figure supplement 1C*).

To classify pSMAD2/3 nuclear accumulation in blastoderm explants and embryos at 50% epiboly as *present*, *strongly reduced* or *absent*, we manually counted the number of bright pSMAD2/3 positive nuclei using Imaris. Wildtype blastoderm explants and embryos had an average of ±22 or ±81 very bright pSMAD2/3 positive nuclei, respectively. A *strongly reduced* pSMAD2/3 domain was defined as a 50% reduction in the number of bright pSMAD2/3 nuclei as compared to the average number detected at this stage in control explants and embryos. Samples were defined as negative when no pSMAD2/3 nuclei could be detected.

## Analysis of cell clone dispersal

To analyze cell clone dispersal, explants expressing Membrane-RFP in all cells and H2A-chFP in cell clones were imaged on a Zeiss LSM 880 upright confocal microscope equipped with a Zeiss Plan-Apochromat 20x/1.0 water immersion objective. The Imaris Spot detection plugin was used to automatically determine the 3D coordinates of all H2A-chFP-expressing nuclei at the start of acquisition (typically corresponding to late sphere stage) and after 1 and 2 hr. As proxies for cell dispersion, both the distance between all labelled nuclei (averaged over all pairs of cells), and the average distance between each cell and the center of mass of the labelled cell cluster were computed for each time point.

## Analysis of nuclear β-catenin positive nuclei

Non-dividing β-catenin positive nuclei were manually selected using the Imaris Spot detection plugin based on their nuclear β-catenin signal. Projection and calculation of distances were performed as described for pSMAD2/3 nuclear immunostainings. To plot the number of β-catenin positive nuclei as a function of the distance to the wounding site, expressed as cell tiers, we measure the diameter of 10 randomly selected cells from each sample using the Line tool using Fiji as described above.

To analyze the clustering of β-catenin positive nuclei, the angular distribution of all positive nuclei in reference to an axis drawn between the wounding site and the brightest β-catenin nuclei was automatically calculated (note that these nuclei were then excluded from the subsequent angle analysis), using the same MATLAB script described for analysis of pSMAD2/3 immunostainings (*Source code 1*).

## Analysis of the *sebox::EGFP*-expressing domain

To assess the extent of mesendoderm induction, as assayed by the pan-mesendodermal marker *sebox*, the same number of z planes were MAX projected in explants of different experimental conditions (Nodal inhibitor treatments, DD-removed and animal pole explants) and corresponding controls using Fiji. To compare the size of the mesendodermal domain, the freehand tool was used to manually outline the *sebox::EGFP*-positive area. For this analysis, only explants, where the complete *sebox::EGFP* was detectable, were included. Based on this selection, explants with a domain 50% smaller than the average area in control explants were categorized as *reduced* mesendoderm induction. Explants, which did not show any obvious EGFP accumulation, were considered to be negative, and the corresponding area of the *sebox::EGFP* domain was considered to be 0.

## Statistics

The number of embryos, blastoderm explants or blastoderm reaggregates analyzed (n) and the number of experimental replicates (N) are indicated in the figure legends. No statistical tests were used to assess sample sizes. No inclusion/exclusion criteria were used and all analyzed samples were included in the analysis. In the graphs, the error bars shown correspond to ± s.d., except in *Figure 1E* and *Figure 1—figure supplement 1I* where the error bars denote ± s.e.m. In box plots, the error bars correspond to the minimum and maximum of the dataset distribution. The statistical test used to assess significance is stated in the figure legends and was chosen after the distribution of each group was tested using the Shapiro-Wilk normality test. To compare two groups, we used either a two-tailed Student *t* test or a two-tailed Mann-Whitney test, depending on whether the dataset shows normal distribution. When using the *t* test, the variances were assessed using the F-test. To compare more than two groups, we used either ANOVA or Kruskal-Wallis test, depending on whether the dataset shows normal distribution. In these cases, a correction for

multiple comparisons was used to increase statistical power. All statistical analyses were performed using GraphPad Prism.

# Additional information

## Funding

| Funder | Grant reference number | Author |
|--------|------------------------|--------|
| H2020 European Research Council | MECSPEC742573 | Carl-Philipp Heisenberg |
| Austrian Academy of Sciences | | Alexandra Schauer |
| European Molecular Biology Organization | 850-2017 | Diana Pinheiro |
| Human Frontier Science Program | LT000429/2018-L2 | Diana Pinheiro |

The funders had no role in study design, data collection and interpretation, or the decision to submit the work for publication.

## Author contributions

Alexandra Schauer, Diana Pinheiro, Conceptualization, Formal analysis, Investigation, Visualization, Methodology, Writing - original draft, Writing - review and editing; Robert Hauschild, Software, Formal analysis; Carl-Philipp Heisenberg, Conceptualization, Resources, Supervision, Funding acquisition, Writing - original draft, Project administration, Writing - review and editing

## Author ORCIDs

Alexandra Schauer ⓘ https://orcid.org/0000-0001-7659-9142
Diana Pinheiro ⓘ https://orcid.org/0000-0003-4333-7503
Robert Hauschild ⓘ http://orcid.org/0000-0001-9843-3522
Carl-Philipp Heisenberg ⓘ https://orcid.org/0000-0002-0912-4566

## Ethics

Animal experimentation: All animal experiments in this study were performed in strict accordance with the guidelines of the Ethics and Animal Welfare Committee (ETK) in Austria. The respective approval number that covers the performed experiments is 66.018/0010-WF/II/3b/2014.

## Decision letter and Author response

Decision letter https://doi.org/10.7554/eLife.55190.sa1
Author response https://doi.org/10.7554/eLife.55190.sa2

# Additional files

## Supplementary files

• Source code 1. pSMAD2/3 and β-catenin analysis.
• Transparent reporting form

## Data availability

All data generated and analyzed in the manuscript are provided as Source data Files. The Custom Script used has been uploaded as Source code 1.

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
