## [Decision Letter]

**Acceptance summary:**

Work from embryonic stem cell cultures has shown that these cells have the capacity to self-organize when isolated from extraembryonic signals, calling into question the essential role of these signals for normal development. In this paper, a zebrafish explant system is used to investigate embryonic patterning in isolation from extraembryonic tissue. Although explants were able to extend and generate a largely complete mesendoderm anlage, their capacity to organize was found to rely on intrinsic genetic programs maternally loaded into the egg and thus were not the result of true self organization. These findings contribute new insights in the role of instructive signals from pre-patterns in self-organization.

**Decision letter after peer review:**

Thank you for submitting your article "Zebrafish embryonic explants undergo genetically encoded self-assembly" for consideration by *eLife*. Your article has been reviewed by two peer reviewers, and the evaluation has been overseen by a Reviewing Editor and Didier Stainier as the Senior Editor. The following individual involved in review of your submission has agreed to reveal their identity: Florence L. Marlow (Reviewer #1).

The reviewers have discussed the reviews with one another and the Reviewing Editor has drafted this decision to help you prepare a revised submission.

Work from embryonic stem cell cultures has shown that these cells have the capacity to self-organize when isolated from extraembryonic signals, which normally contribute to pattern formation in vivo. These findings call into question the essential role of extraembryonic signals for normal development. In the zebrafish embryo, maternal dorsal determinants as well as the later forming extraembryonic yolk syncytial layer have both been shown to contribute to embryonic germ layer specification and patterning. In this work, Schauer et al., use a zebrafish explant system to investigate embryonic patterning in isolation from the yolk cell. Explants formed by isolation of the embryonic blastoderm at the 256-cell stage formed polarized patterns of mesendodermal and ectodermal markers when cultured in a simple saline medium. The authors provide evidence that dorsal determinants transported from the yolk elicited β-catenin activation and subsequently polarized Nodal signaling in the explants. Following this initial polarization, the explants elongated to form embryoid-like structures in a non-canonical Wnt/PCP signaling-dependent manner. Although some explants were able to generate a largely complete mesendoderm anlage, their capacity to organize is not true self organization because it relies on intrinsic genetic programs maternally loaded into the egg. In addition, signals from the yolk syncytial layer are required for high and stable signaling to achieve robust germ layer specification in vivo.

Overall the reviewers were enthusiastic about the work and agreed that it is of broad interest and makes a new contribution to understanding of the role of instructive signals from pre-patterns in self-organization.

Essential revisions:

1) To better assess how well the explant system recapitulates normal development – in particular the spatial relationships between ectoderm, mesoderm, and endoderm – and to directly support the model shown in Figure 1G, the authors should provide co-stainings. Double *in situ* hybridizations for *ntl/sox17, ntl/gata2, ntl/papc* (or *myod*) and *ntl/gsc* (or *hgg*), would help to better characterize the germ layer relationships (i.e. keeping *ntl* as a constant reference point).

2) In Figure 2B the percent of SMAD positive nuclei seems to go down in the embryo over time but goes up in the explants. This is also the case, in Figure 2C where the intensity goes up in explants but down in embryos. Could there be a delay in zygotic genome activation in the explants? It seems possible that the explants that do not elongate may fail to activate the genome and thus do not progress. To address this point the authors can examine the timing of expression of some early ZGA markers in explants compared to embryos (for examples, see markers suggested in point 4 below).

3) The explants seem to have a variety of shapes and sizes in the various experiments and to have variably sized cavities. Did the authors examine proliferation in the explants? Are the differences in size due to proliferation differences or differences in compaction of the explants? This is interesting in light of the different contributions of proliferation to zebrafish gastrulation, where it seems to be important to have enough cells to initiate genome activation and thus gastrulation but does not contribute to gastrulation/morphogenesis in a significant manner. This is in contrast to other systems, like mouse where proliferation seems to play a bigger role. Can the authors comment on this?

4) Mouse gastruloids display dorsal-ventral polarity (Beccari et al., 2018). Is this also the case for zebrafish explants? Please assess the expression patterns of *bmp2* and chordin as well as BMP signaling (pSMAD1/5 staining). This is particularly important for comparison to *Xenopus* experiments, in which the wound closes in a manner similar to the zebrafish explants, juxtaposing dorsal and ventral signaling centers (Moriyama and De Robertis PNAS 2018). Does the organizer shift in a similar manner in zebrafish as in *Xenopus*? Please also explain how the wounding site was identified in the images (e.g. in Figure 2A, shield stage).

5) The analysis of the experiment removing dorsal determinants suggests that axis extension is only marginally affected by removal of dorsal determinants, which is unexpected. Therefore, please add an intact control embryo at 1 dpf to show that the removal of dorsal determinants indeed gives the expected phenotype. Furthermore, please quantify the pSMAD2/3 stainings in Figure 4—figure supplement 2A and show the expression patterns of mesendodermal markers in the corresponding explants.

6) Since convergence and extension regulators are examined and given the various shapes of the extensions, it would be informative to also measure the width or thickness of the extension or area to account for overall growth/shape tissue change.

7) Figure 4D requires improved quantification. This is particularly necessary for the animal pole explants, which have different cell spacing and a very different punctate pSMAD2/3 staining compared to the other samples. One suggestion is to show all pSMAD2/3 quantifications as clearly as in Figure 5F and G. Or this might be addressed by more clearly defining the cutoff for strongly reduced – is this a 50% reduction etc. Furthermore, background subtraction should be performed in the same way for all samples (i.e. shield stage explants should not be treated differently as described in the second paragraph of the subsection “Analysis of pSMAD2/3 nuclear accumulation”).

8) Please clarify why the *gsc* expression domain appears to be increased in *sqt*+*cyc* MO injected embryos in Figure 5—figure supplement 1C, since the opposite is expected. Suggestions would be to repeat the experiment or provide further quantification.

9) Referring to the explant extensions as "tail-like" seems inaccurate given that the extensions are first apparent at the embryo equivalent of shield stage, several hours prior to tailbud formation. Without experimental support that they are "tail-like" they could be more simply referred to as 'extensions'.

10) Could the authors clarify which cells in the explants are expressing the YSL specific genes shown in Figure 1—figure supplement 3. Do the cavities have nuclei in them-and how do the cavities relate to the cut sites?

Minor point:

In the Discussion, the authors refer to a preprint manuscript by Trivedi et al, 2019. Because there is an apparent discrepancy in the findings, it would seem that addition of serum would be a straight-forward experiment to resolve the different outcomes observed in the two studies. Otherwise, the authors might consider removing this reference to work that has not yet undergone peer review.

---

## [Author Response]

Essential revisions:1) To better assess how well the explant system recapitulates normal development – in particular the spatial relationships between ectoderm, mesoderm, and endoderm – and to directly support the model shown in Figure 1G, the authors should provide co-stainings. Double *in situ* hybridizations for ntl/sox17, ntl/gata2, ntl/papc (or myod) and ntl/gsc (or hgg), would help to better characterize the germ layer relationships (i.e. keeping ntl as a constant reference point).

We have now performed double *in situ* hybridizations to confirm the relative distribution of the different progenitor cell types within bud stage blastoderm explants. Generally, this analysis confirmed the distribution schematized in Figure 1G, and these new data are now provided in Figure 1—figure supplement 3.

2) In Figure 2B the percent of SMAD positive nuclei seems to go down in the embryo over time but goes up in the explants. This is also the case, in Figure 2C where the intensity goes up in explants but down in embryos. Could there be a delay in zygotic genome activation in the explants? It seems possible that the explants that do not elongate may fail to activate the genome and thus do not progress. To address this point the authors can examine the timing of expression of some early ZGA markers in explants compared to embryos (for examples, see markers suggested in point 4 below).

We have now examined the onset of ZGA in blastoderm explants and stage-matched embryos by analyzing the expression of *chrd* and *tbx16*, two genes known to be expressed upon activation of the zygotic genome (Gagnon et al., 2018; Lee et al., 2013). This analysis revealed no systematic differences in ZGA onset between embryos and explants (Figure 1—figure supplement 1D), arguing against the idea that the different temporal dynamics of pSMAD2/3 nuclear accumulation in blastoderm explants or the different degrees of explant extension are due to differences in ZGA onset. Consistent with this, we have also found that nuclear pSMAD2/3, a hallmark of Nodal signaling activation, can be detected as early as high stage in both blastoderm explants and stage-matched embryos (Figure 2—figure supplement 1C). This suggests that the build-up of Nodal signaling is indeed slower in blastoderm explants as compared to stage-matched embryos and not due to delayed ZGA in explants. Finally, we would like to mention that for limiting the developmental variability among blastoderm explants, we have systematically excluded explants with delayed cell cleavages, as assessed by increased cell size approximately 1 hour after explant preparation, from our analysis.

3) The explants seem to have a variety of shapes and sizes in the various experiments and to have variably sized cavities. Did the authors examine proliferation in the explants? Are the differences in size due to proliferation differences or differences in compaction of the explants? This is interesting in light of the different contributions of proliferation to zebrafish gastrulation, where it seems to be important to have enough cells to initiate genome activation and thus gastrulation but does not contribute to gastrulation/morphogenesis in a significant manner. This is in contrast to other systems, like mouse where proliferation seems to play a bigger role. Can the authors comment on this?

We have now analyzed cell size as a readout for cell cleavages in blastoderm explants and stage-matched embryos. This analysis revealed no significant differences in cell size between explants and embryos from high to shield stage (Figure 1—figure supplement 1C), suggesting that variations in cell proliferation are unlikely to underly the observed differences in explant size. Consistent with this, we also found no striking differences in the ability of explants to form a normal-sized extension, when cell proliferation was blocked from shield to bud stage, the period when blastoderm explants typically undergo extension (Figure 3—figure supplement 3D-F). We have also now mentioned the contribution of cell proliferation to body axis elongation in different systems in the Discussion section of the revised manuscript.

As to the potential role of compaction in explant size variation, we have attempted to analyze nuclear density in different regions of the explant; however, this analysis showed considerable differences depending on the different regions of the explants considered and different stages of maturation, precluding a meaningful comparative quantification. We currently favor a model whereby the amount of specified mesendoderm critically affects explant extension (see also our response to point 5 below).

4) Mouse gastruloids display dorsal-ventral polarity (Beccari et al., 2018). Is this also the case for zebrafish explants? Please assess the expression patterns of bmp2 and chordin as well as BMP signaling (pSMAD1/5 staining). This is particularly important for comparison to *Xenopus* experiments, in which the wound closes in a manner similar to the zebrafish explants, juxtaposing dorsal and ventral signaling centers (Moriyama and De Robertis PNAS 2018). Does the organizer shift in a similar manner in zebrafish as in *Xenopus*? Please also explain how the wounding site was identified in the images (e.g. in Figure 2A, shield stage).

We have now analyzed the expression of the dorsoventral marker genes *chrd* and *bmp2b* in blastoderm explants by *in situ* hybridization. This analysis revealed that blastoderm explants are indeed polarized along their dorsoventral (DV) axis (Figure 2—figure supplement 2E). Moreover, we detected a gradient of nuclear pSMAD1/5, as readout of BMP signaling activity along the DV axis (Pomreinke et al., 2017, Zinski et al., 2017), in both blastoderm explants and embryos at 50% epiboly (Figure 2—figure supplement 2F), further supporting the notion of DV explant polarization.

Regarding a potential shift in organizer localization following explant preparation, as observed in frogs, we were – mainly for technical reasons – unable to address this question. While in *Xenopus* embryos, the first cleavage furrow demarcates the future DV axis (Klein, 1987), in zebrafish, the DV axis is not fixed in respect to the early cleavage planes (Helde et al., 1994, Abdelilah et al., 1994), thereby precluding a similar labelling of the future dorsal and ventral halves of the embryo needed for addressing a potential organizer shift following explant preparation.

Finally, the wounding site in the blastoderm explants is identified based on a combination of features, namely the increased and irregular junctional staining of β-catenin and/or comparatively lower and more irregular nuclei density. The criteria used to define the wounding site are included in the corresponding section of Materials and methods.

5) The analysis of the experiment removing dorsal determinants suggests that axis extension is only marginally affected by removal of dorsal determinants, which is unexpected. Therefore, please add an intact control embryo at 1 dpf to show that the removal of dorsal determinants indeed gives the expected phenotype. Furthermore, please quantify the pSMAD2/3 stainings in Figure 4—figure supplement 2A and show the expression patterns of mesendodermal markers in the corresponding explants.

We thank the reviewers for this insightful comment. We have now analyzed the morphology of DD-removed embryos at 1 dpf and found that the majority of DD-removed embryos indeed displayed strong ventralization. Yet, a considerable portion of these embryos (around 40%) still formed a rudimentary dorsal axis or even seemingly complete head structures (Figure 4—figure supplement 1A, B). This finding indicates that DD-removal at the 1 cell-stage has variable effects on embryo DV patterning, which might also explain why explant extension in DD-removed embryos is not more drastically affected (Figure 4D-I).

We have also found strongly reduced, yet variable, mesendoderm induction in DD-removed explants (Figure 4—figure supplement 1E-G) and further observed that the amount of explant extension closely correlated with the size of the *sebox::EGFP* domain present therein (Figure 4—figure supplement 1E, H). A similar relationship could be observed upon Nodal signaling inhibition at different stages of explant maturation (please see Figure 2E-H and Figure 2—figure supplement 3A-D) and in animal pole explants (Figure 4F-H and Figure 4—figure supplement 1E-H). Overall, these data suggest that variability in mesendoderm induction might underly the variety of shapes of blastoderm explants by the end of gastrulation.

Finally, we have now quantified the pSMAD2/3 stainings shown in Figure 4—figure supplement 2A. Consistent with a partially redundant role of dorsal determinants and the YSL in boosting Nodal signaling within the overlying blastoderm, we found that control and DD-removed embryos showed similar percentages of pSMAD2/3 positive nuclei and similar intensities of the brightest pSMAD2/3 positive nuclei (Figure 4—figure supplement 2B-D). Moreover, mesendoderm induction, as determined by expression of the panmesendodermal marker *sebox::EGFP*, still occurred in DD-removed embryos (Figure 4—figure supplement 2E, F).

6) Since convergence and extension regulators are examined and given the various shapes of the extensions, it would be informative to also measure the width or thickness of the extension or area to account for overall growth/shape tissue change.

We fully agree with the reviewer’s comment and have therefore now included a descriptive analysis of how explant area, width and extension length change over time. In line with our previous findings, we observed that the explant area increased over time, and that the length of the extended explant portion increased while the width of the non-extended portion decreased (Figure 3—figure supplement 1A-C).

7) Figure 4D requires improved quantification. This is particularly necessary for the animal pole explants, which have different cell spacing and a very different punctate pSMAD2/3 staining compared to the other samples. One suggestion is to show all pSMAD2/3 quantifications as clearly as in Figure 5F and G. Or this might be addressed by more clearly defining the cutoff for strongly reduced – is this a 50% reduction etc. Furthermore, background subtraction should be performed in the same way for all samples (i.e. shield stage explants should not be treated differently as described in the second paragraph of the subsection “Analysis of pSMAD2/3 nuclear accumulation”).

Since the nuclear accumulation of pSMAD2/3 is much lower in DD-removed or animal pole explants compared to explants prepared from wildtype embryos, the cut-off level between positive and negative nuclei in those experimental conditions is much more difficult to determine. Moreover, our analysis in Figures 2, 5 and 6 is restricted to the nuclei considered to be positive for pSMAD2/3 (see Materials and methods for additional details). To address the reviewer’s comment, we have therefore now clarified in the corresponding Materials and methods section the cutoff criteria used to classify a sample as exhibiting strongly reduced pSMAD2/3 signaling. In short, a strongly reduced pSMAD2/3 domain in 50% epiboly blastoderm explants was defined as having less than 12 clearly distinguishable pSMAD2/3 positive nuclei, a reduction of roughly 50% compared to the average number of brightest positive nuclei at this stage in control explants (β 22 nuclei; see Figure 2A). Correspondingly, the threshold for a strongly reduced pSMAD2/3 domain in embryos at 50% epiboly has been set at 40 clearly distinguishable pSMAD2/3 positive nuclei ( ± 81 nuclei in control embryos, see Figure 2A). To better illustrate this, we have also provided a video with the high-resolution images of pSMAD2/3 in control, DD-removed and animal pole explants (Video 4).

Finally, we have now performed background subtraction in the same way for all analyzed samples and updated the graphs in Figure 2 and Figure 2—figure supplement 1.

8) Please clarify why the gsc expression domain appears to be increased in sqt+cyc MO injected embryos in Figure 5—figure supplement 1C, since the opposite is expected. Suggestions would be to repeat the experiment or provide further quantification.

We thank the reviewers for this comment. Indeed, the *gsc::EGFP-CAAX* signal in the control MO YSL-injected embryos (dorsal view) is particularly low, suggesting that the *sqt* + *cyc* MO YSL-injected embryos is increased. Thus, we have now exchanged this image for a more representative example.

In addition to this, we would like to note that the Tg(*gsc::EGFP-CAAX*) line is leaky, and EGFP expression can be detected in both the prechordal plate and, to a lesser extent, also in the notochord (Author response image 1), which is still induced in *sqt* + *cyc* MO YSL-injected embryos (Figure 5A, Figure 5—figure supplement 1B). This might explain why in some samples, *gsc::EGFP-CAAX* expression appears similar or even increased in *sqt* + *cyc* MO YSL-injected embryos as compared to control MO YSL-injected embryos.

**Author response image 1. respfig1:** Expression of the YSL marker gene *mxtx2* in blastoderm explants, prepared from high stage embryos. (**A**) Exemplary high-resolution fluorescence images of a *gsc::GFP-CAAX*-expressing embryo (green) at different stages of gastrulation. The cell nuclei (grey) are marked by the expression of H2A-chFP. Dorsal side views. (**B**) Expression of the YSL marker gene *mxtx2*, as determined by whole mount *in situ* hybridization in 50% epiboly stage embryos (side view) and blastoderm explants (side view) that were prepared from high stage embryos. The proportion of embryos or blastoderm explants with a phenotype similar to the images shown is indicated in the lower right corner (embryos: n=19, N=3; explants: n=19, N=3). Scale bars: 100 µm (**A**), 200 µm (**B**).

9) Referring to the explant extensions as "tail-like" seems inaccurate given that the extensions are first apparent at the embryo equivalent of shield stage, several hours prior to tailbud formation. Without experimental support that they are "tail-like" they could be more simply referred to as 'extensions'.

We have now removed the term "tail-like" extension and refer to the extended portion of the blastoderm explants simply as “extension” to avoid any confusion.

10) Could the authors clarify which cells in the explants are expressing the YSL specific genes shown in Figure 1—figure supplement 3. Do the cavities have nuclei in them-and how do the cavities relate to the cut sites?

It is difficult to unambiguously determine which cells within blastoderm explants can still express some YSL markers. One straightforward possibility would be that these cells originate from incomplete removal of YSL progenitor cells during explant preparation. Additionally, some YSL markers, such as *mxtx2*, have been shown to be expressed also in marginal cells of the blastoderm (Hirata et al., 2000), suggesting that the *mxtx2* positive cells within the explant might potentially originate from these marginal cells. Supporting the possibility that *mxtx2* positive cells within the explant originate from YSL progenitor cells present within the blastoderm, we found that *mxtx2* expression was lost when the blastoderm explants are prepared at high stage, when the YSL has already formed and thus the progenitor cells are absent (Author response image 1).

To illustrate the spatial relationship between cavity formation and the original cutting site, we have now included Video 2 showing how the interstitial fluid is uniformly distributed throughout the explant, prior to the formation of a central cavity opposite to the extended portion of the explant. Moreover, this video shows that while no coordinated movement of cells into the cavity can be observed, sporadic cells can be found within the cavity.

In the Discussion, the authors refer to a preprint manuscript by Trivedi et al, 2019. Because there is an apparent discrepancy in the findings, it would seem that addition of serum would be a straight-forward experiment to resolve the different outcomes observed in the two studies. Otherwise, the authors might consider removing this reference to work that has not yet undergone peer review.

Following the reviewer’s suggestion, we have now removed this sentence and the corresponding reference.

[Editors’ note: following publication and feedback from the community, this reference was reinstated.]